# Semi-Supervised Semantic Segmentation via Marginal Contextual Information

**Moshe Kimhi**                                                          *moshekimhi@cs.technion.ac.il*
*Computer Science Department, Technion*

**Shai Kimhi**                                                           *shaikimhi221199@gmail.com*
*Computer Science Department, Technion*

**Evgenii Zheltonozhskii**                                              *evgeniizh@campus.technion.ac.il*
*Physics Department, Technion*

**Or Litany**                                                           *orlitany@technion.ac.il*
*Computer Science Department, Technion*
*NVIDIA*

**Chaim Baskin**                                                        *chaimbaskin@technion.ac.il*
*Computer Science Department, Technion*

**Reviewed on OpenReview:** *https://openreview.net/forum?id=i5yKW1pmjW*

## Abstract

We present a novel confidence refinement scheme that enhances pseudo labels in semi-supervised semantic segmentation. Unlike existing methods, which filter pixels with low-confidence predictions in isolation, our approach leverages the spatial correlation of labels in segmentation maps by grouping neighboring pixels and considering their pseudo labels collectively. With this contextual information, our method, named S4MC, increases the amount of unlabeled data used during training while maintaining the quality of the pseudo labels, all with negligible computational overhead. Through extensive experiments on standard benchmarks, we demonstrate that S4MC outperforms existing state-of-the-art semi-supervised learning approaches, offering a promising solution for reducing the cost of acquiring dense annotations. For example, S4MC achieves a 1.39 mIoU improvement over the prior art on PASCAL VOC 12 with 366 annotated images. The code to reproduce our experiments is available at https://s4mcontext.github.io/.

## 1 Introduction

Supervised learning has been the driving force behind advancements in modern computer vision, including classification (Krizhevsky et al., 2012; Dai et al., 2021), object detection (Girshick, 2015; Zong et al., 2022), and segmentation (Zagoruyko et al., 2016; Chen et al., 2018a; Li et al., 2022; Kirillov et al., 2023). However, it requires extensive amounts of labeled data, which can be costly and time-consuming to obtain. In many practical scenarios, there is no shortage of available data, but only a fraction can be labeled due to resource constraints. This challenge has led to the development of semi-supervised learning (SSL; Rasmus et al., 2015; Berthelot et al., 2019; Sohn et al., 2020a; Yang et al., 2022a), a methodology that leverages both labeled and unlabeled data for model training.

This paper focuses on applying SSL to semantic segmentation, which has applications in various areas such as perception for autonomous vehicles (Bartolomei et al., 2020), mapping (Van Etten et al., 2018) and

**Before refinement**     **After refinement**

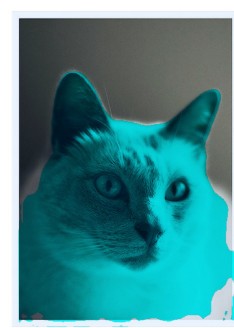     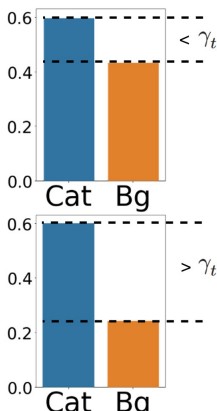

Figure 1: **Confidence refinement** observation over one class (Cat). **Left:** pseudo labels generated without refinement. **Middle:** pseudo labels obtained from the same model after refinement with marginal contextual information. **Right Top:** predicted probabilities of the top two classes of the pixel highlighted by the red square before, and **Bottom:** after refinement. S4MC allows additional correct pseudo labels to propagate.

agriculture (Milioto et al., 2018). SSL is particularly appealing for segmentation tasks, as manual labeling can be prohibitively expensive.

A widely adopted approach for SSL is pseudo labeling (Lee, 2013). This technique dynamically assigns supervision targets to unlabeled data during training based on the model's predictions. To generate a meaningful training signal, it is essential to adapt the predictions before integrating them into the learning process. Several techniques have been proposed, such as using a teacher network to generate supervision to a student network (Tarvainen & Valpola, 2017; Ke et al., 2019; Cai et al., 2022). Additionally, the teacher may undergo weaker augmentations than the student (Berthelot et al., 2019), simplifying the teacher's task.

However, pseudo labeling is intrinsically susceptible to confirmation bias, which tends to reinforce the model predictions instead of improving the student model. Mitigating confirmation bias becomes particularly important when dealing with erroneous predictions made by the teacher network.

Confidence-based filtering is a popular technique to address this issue (Sohn et al., 2020a). This approach assigns pseudo labels only when the model's confidence surpasses a specified threshold, reducing the number of incorrect pseudo labels. Though simple, this strategy was proven effective and inspired multiple improvements in semi-supervised classification (Zhang et al., 2021; Rizve et al., 2021), segmentation (Wang et al., 2022), and object detection (Sohn et al., 2020b; Liu et al., 2021; Zhao et al., 2020; Wang et al., 2021). However, the strict filtering of the supervision signal leads to extended training periods and, potentially, to overfitting when the labeled instances are insufficient to represent the entire sample distribution. Lowering the threshold would allow for higher training volumes at the cost of reduced quality, further hindering the performance (Sohn et al., 2020a).

In response to these challenges, we introduce a novel confidence refinement scheme for the teacher network predictions in segmentation tasks designed to increase the availability of pseudo labels without sacrificing their accuracy. Drawing on the observation that labels in segmentation maps exhibit strong spatial correlation, we propose to group neighboring pixels and collectively consider their pseudo labels. When considering pixels in spatial groups, we asses the event-union probability, which is the probability that at least one pixel belongs to a given class. We assign a pseudo label if this probability is sufficiently larger than the event-union probability of any other class. By taking context into account, our approach *Semi-Supervised Semantic Segmentation via Marginal Contextual Information* (S4MC), enables a relaxed filtering criterion which increases the number of unlabeled pixels utilized for learning while maintaining high-quality labeling, as demonstrated in Fig. 1.

We evaluated S4MC on multiple benchmarks. S4MC achieves significant improvements in performance over previous state-of-the-art methods. In particular, we observed an increase of **+1.39 mIoU** on PASCAL VOC 12 (Everingham et al., 2010) using 366 annotated images, **+1.01 mIoU** on Cityscapes (Cordts et al., 2016)

using only 186 annotated images, and increase **+1.5 mIoU** on COCO (Lin et al., 2014) using 463 annotated images. These findings highlight the effectiveness of S4MC in producing high-quality segmentation results with minimal labeled data.

## 2 Related Work

### 2.1 Semi-Supervised Learning

Pseudo labeling (Lee, 2013) is an effective technique in SSL, where labels are assigned to unlabeled data based on model predictions. To make the most of these labels during training, it is essential to refine them (Laine & Aila, 2016; Berthelot et al., 2019; 2020; Xie et al., 2020). This can be done through consistency regularization (Laine & Aila, 2016; Tarvainen & Valpola, 2017; Miyato et al., 2018), which ensures consistent predictions between different views or different models' prediction of the unlabeled data. To ensure that the pseudo labels are helpful, the temperature of the prediction (soft pseudo labels; Berthelot et al., 2019) can be increased, or the label can be assigned to samples with high confidence (hard pseudo labels; Xie et al., 2020; Sohn et al., 2020a; Zhang et al., 2021).

### 2.2 Semi-Supervised Semantic Segmentation

In semantic segmentation, most SSL methods rely on consistency regularization and developing augmentation strategies compatible with segmentation tasks (French et al., 2020; Ke et al., 2020; Chen et al., 2021; Zhong et al., 2021; Xu et al., 2022). Given the uneven distribution of labels typically encountered in segmentation maps, techniques such as adaptive sampling, augmentation, and loss re-weighting are commonly employed (Hu et al., 2021). Feature perturbations (FP) on unlabeled data (Ouali et al., 2020; Zou et al., 2021; Liu et al., 2022b; Yang et al., 2023) are also used to enhance consistency and the virtual adversarial training (Liu et al., 2022b). Curriculum learning strategies that incrementally increase the proportion of data used over time are beneficial in exploiting more unlabeled data (Yang et al., 2022b; Wang et al., 2022). A recent approach introduced by Wang et al. (2022) included unreliable pseudo labels into training by employing contrastive loss with the least confident classes predicted by the model. Unimatch (Yang et al., 2023) combined SSL (Sohn et al., 2020a) with several self-supervision signals, i.e., two strong augmentations and one more with FP, obtained good results without complex losses or class-level heuristics. However, most existing works primarily focus on individual pixel label predictions. In contrast, we delve into the contextual information offered by spatial predictions on unlabeled data.

### 2.3 Contextual Information

Contextual information encompasses environmental cues that assist in interpreting and extracting meaningful insights from visual perception (Toussaint, 1978; Elliman & Lancaster, 1990). Incorporating spatial context explicitly has been proven beneficial in segmentation tasks, for example, by encouraging smoothness like in the Conditional Random Fields method (Chen et al., 2018a) and attention mechanisms (Vaswani et al., 2017; Dosovitskiy et al., 2021; Wang et al., 2020). Combating dependence on context has shown to be helpful by Nekrasov et al. (2021). This work uses the context from neighboring pixel predictions to enhance pseudo label propagation.

## 3 Method

This section describes the proposed method using the teacher–student paradigm with teacher averaging (Tarvainen & Valpola, 2017). Adjustments for image-level consistency are described in Appendix F.

### 3.1 Overview

In semi-supervised semantic segmentation, we are given a labeled training set $\mathcal{D}_\ell = \left\{ (\mathbf{x}_i^\ell, \mathbf{y}_i) \right\}_{i=1}^{N_\ell}$, and an unlabeled set $\mathcal{D}_u = \{\mathbf{x}_i^u\}_{i=1}^{N_u}$ sampled from the same distribution, i.e., $\left\{ \mathbf{x}_i^\ell, \mathbf{x}_i^u \right\} \sim D_x$. Here, $\mathbf{y}$ are 2D tensors

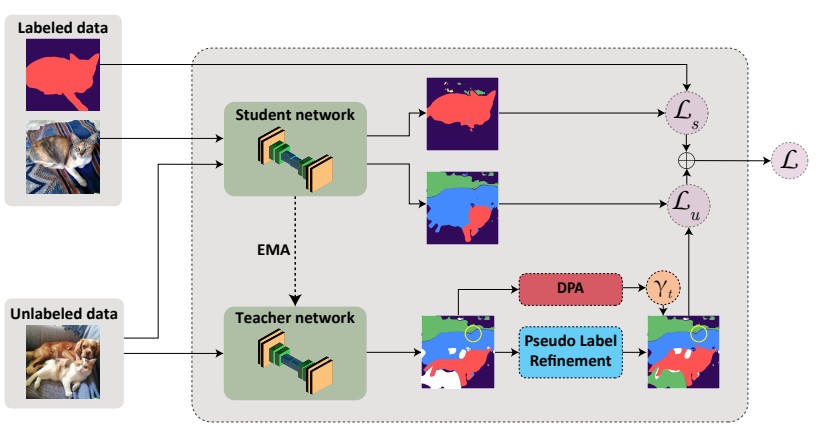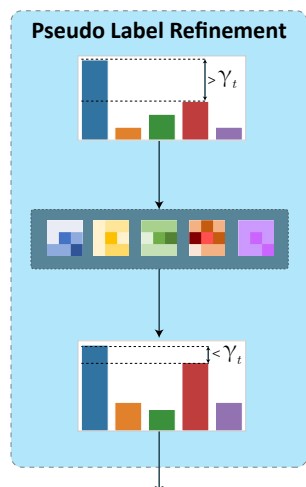

Figure 2: **Left:** S4MC employs a teacher–student paradigm for semi-supervised segmentation. Labeled images are used to supervise the student network directly; both networks process unlabeled images. Teacher predictions are refined and used to evaluate the margin value, which is then thresholded to produce pseudo labels that guide the student network. The threshold, denoted as $\gamma_t$, is dynamically adjusted based on the teacher network's predictions. **Right:** Our confidence refinement module exploits neighboring pixels to adjust per-class predictions, as detailed in Section 3.2.1. The class distribution of the pixel marked by the yellow circle on the left is changed. Before refinement, the margin surpasses the threshold and erroneously assigns the blue class (dog) as a pseudo label. After refinement, the margin reduces, thereby preventing error propagation.

of shape $H \times W$, assigning a semantic label to each pixel of $\mathbf{x}$. We aim to train a neural network $f_\theta$ to predict the semantic segmentation of unseen images sampled from $D_x$.

We follow a teacher-averaging approach and train two networks $f_{\theta_s}$ and $f_{\theta_t}$ that share the same architecture but update their parameters separately. The student network $f_{\theta_s}$ is trained using supervision from the labeled samples and pseudo labels created by the teacher's predictions for unlabeled ones. The teacher model $f_{\theta_t}$ is updated as an exponential moving average (EMA) of the student weights. $f_{\theta_s}(\mathbf{x}_i)$ and $f_{\theta_t}(\mathbf{x}_i)$ denote the predictions of the student and teacher models for the $\mathbf{x}_i$ sample, respectively. At each training step, a batch of $\mathcal{B}_\ell$ and $\mathcal{B}_u$ images is sampled from $\mathcal{D}_\ell$ and $\mathcal{D}_u$, respectively. The optimization objective can be written as the following loss:

$$\mathcal{L} = \mathcal{L}_s + \lambda \mathcal{L}_u \tag{1}$$

$$\mathcal{L}_s = \frac{1}{M_l} \sum_{\mathbf{x}_i^\ell, \mathbf{y}_i \in \mathcal{B}_l} \ell_{CE}(f_{\theta_s}(\mathbf{x}_i^\ell), \mathbf{y}_i) \tag{2}$$

$$\mathcal{L}_u = \frac{1}{M_u} \sum_{\mathbf{x}_i^u \in \mathcal{B}_u} \ell_{CE}(f_{\theta_s}(\mathbf{x}_i^u), \hat{\mathbf{y}}_i), \tag{3}$$

where $\mathcal{L}_s$ and $\mathcal{L}_u$ are the losses over the labeled and unlabeled data correspondingly, $\lambda$ is a hyperparameter controlling their relative weight, and $\hat{\mathbf{y}}_i$ is the pseudo label for the $i$-th unlabeled image. Not every pixel of $\mathbf{x}_i$ has a corresponding label or pseudo label, and $M_l$ and $M_u$ denote the number of pixels with label and assigned pseudo label in the image batch, respectively.

### 3.1.1 Pseudo Label Propagation

For a given image $\mathbf{x}_i$, we denote by $x_{j,k}^i$ the pixel in the $j$-th row and $k$-th column. We adopt a thresholding-based criterion inspired by FixMatch (Sohn et al., 2020a). By establishing a score, denoted as $\kappa$, which is based on the class distribution predicted by the teacher network, we assign a pseudo label to a pixel if its

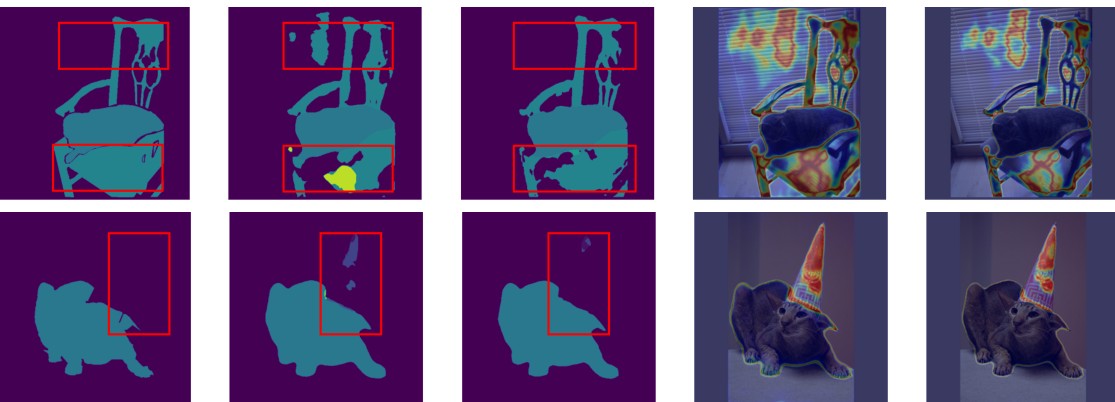

Figure 3: **Qualitative results.** *Segmentation mask* from left to right: Ground Truth, CutMix-Seg and CutMix-Seg+S4MC. *Heat map* left is CutMix-Seg and right CutMix-Seg+S4MC, represents the uncertainty of the model ($\kappa^{-1}$), showing more confident predictions in certain areas and smoother segmentation maps (marked by the red boxes). Additional examples are shown in Appendix A.

score exceeds a threshold $\gamma_t$:

$$\hat{\mathbf{y}}_{j,k}^i = \begin{cases} \arg\max_c\{p_c(x_{j,k}^i)\} & \text{if } \kappa(x_{j,k}^i; \theta_t) > \gamma_t, \\ \text{ignore} & \text{otherwise,} \end{cases} \tag{4}$$

where $p_c(x_{j,k}^i)$ is the pixel probability of class $c$. A commonly used score is given by $\kappa(x_{j,k}^i; \theta_t) = \max_c\{p_c(x_{j,k}^i)\}$. However, using a pixel-wise margin (Scheffer et al., 2001; Shin et al., 2021), produces more stable results. Denoting by max2 the second-highest value, the margin is given by the difference between the highest and the second-highest values of the probability vector:

$$\kappa_{\text{margin}}(x_{j,k}^i) = \max_c\{p_c(x_{j,k}^i)\} - \max2_c\{p_c(x_{j,k}^i)\}, \tag{5}$$

### 3.1.2 Dynamic Partition Adjustment (DPA)

Following Wang et al. (2022), we use a decaying threshold $\gamma_t$. DPA replaces the fixed threshold with a quantile-based threshold that decreases with time. At each iteration, we set $\gamma_t$ as the $\alpha_t$-th quantile of $\kappa_{\text{margin}}$ over all pixels of all images in the batch. $\alpha_t$ linearly decreases from $\alpha_0$ to zero during the training:

$$\alpha_t = \alpha_0(1 - {}^t/_{iterations})$$

As the model predictions improve with each iteration, gradually lowering the threshold increases the number of propagated pseudo labels without compromising quality.

### 3.2 Marginal Contextual Information

Utilizing contextual information (Section 2.3), we look at surrounding predictions (predictions on neighboring pixels) to refine the semantic map at each pixel. We introduce the concept of "Marginal Contextual Information," which involves integrating additional information to enhance predictions across all classes. At the same time, reliability-based pseudo label methods focus on the dominant class only (Sohn et al., 2020a; Wang et al., 2023). Section 3.2.1 describes our confidence refinement, followed by our thresholding strategy and a description of S4MC methodology.

### 3.2.1 Confidence Margin Refinement

We refine each pixel's predicted pseudo label by considering its neighboring pixels' predictions. Given a pixel $x_{j,k}^i$ with a corresponding per-class prediction $p_c(x_{j,k}^i)$, we examine neighboring pixels $x_{\ell,m}^i$ within an $N \times N$

pixel neighborhood surrounding it. As an example for using one neighbor, we then calculate the probability that at least one of the two pixels belongs to class $c$:

$$p_c(x^i_{j,k} \cup x^i_{\ell,m}) = p_c(x^i_{j,k}) + p_c(x^i_{\ell,m}) - p_c(x^i_{j,k}, x^i_{\ell,m}), \tag{6}$$

where $p_c(x^i_{j,k}, x^i_{\ell,m})$ denote the joint probability of both $x^i_{j,k}$ and $x^i_{\ell,m}$ belonging to the same class $c$.

While the model does not predict joint probabilities, assuming a non-negative correlation between the probabilities of neighboring pixels is reasonable. This is mainly due to the nature of segmentation maps, which are typically piecewise constant. The joint probability can thus be bounded from below by assuming independence: $p_c(x^i_{j,k}, x^i_{\ell,m}) \geqslant p_c(x^i_{j,k}) \cdot p_c(x^i_{\ell,m})$. By substituting this into Eq. (6), we obtain an upper bound for the event union probability:

$$p_c(x^i_{j,k} \cup x^i_{\ell,m}) \leq p_c(x^i_{j,k}) + p_c(x^i_{\ell,m}) - p_c(x^i_{j,k}) \cdot p_c(x^i_{\ell,m}). \tag{7}$$

For each class $c$, we select the neighbor with the maximal information utilization using Eq. (7):

$$\tilde{p}_c(x^i_{j,k}) = \max_{\ell,m} p_c(x^i_{j,k} \cup x^i_{\ell,m}). \tag{8}$$

Computing the event union over all classes employs neighboring predictions to amplify differences in ambiguous cases. Similarly, this prediction refinement prevents the creation of over-confident predictions not supported by additional spatial evidence and helps reduce confirmation bias. The refinement is visualized in Fig. 1. In our experiments, we used a neighborhood size of $3 \times 3$. To determine whether the incorporation of contextual information could be enhanced with larger neighborhoods, we conducted an ablation study focusing on the neighborhood size and the neighbor selection criterion, as detailed in Table 7. For larger neighborhoods, we decrease the probability contribution of the neighboring pixels with a distance-dependent factor:

$$\tilde{p}_c(x^i_{j,k}) = p_c(x^i_{j,k}) + \beta_{\ell,m} \big[ p_c(x^i_{\ell,m}) - p_c(x^i_{j,k}, x^i_{\ell,m}) \big], \tag{9}$$

where $\beta_{\ell,m} = \exp\!\big(-\frac{1}{2}(|\ell - j| + |m - k|)\big)$ is a spatial weighting function. Empirically, contextual information refinement affects mainly the most probable one or two classes. This aligns well with our choice to use the margin confidence (5).

We explored alternatives at the pixel level, such as the maximum class probability ($\kappa_{\max}$) and entropy ($\kappa_{\mathrm{ent}}$). Table F.1 in the Appendix studies the impact of different confidence functions on pseudo label refinement.

Considering multiple neighbors, we can use the formulation for three or more events. In practice, we use the associative law of set theory calculate it iteratively by grouping two events at each iteration, and exclude the chosen one from the remaining neighborhood.

### 3.2.2 Threshold Setting

A high threshold can prevent transferring the teacher model's wrong "beliefs" to the student model. However, this comes at the expense of learning from fewer examples, resulting in a less comprehensive model. To determine the DPA threshold, we use the teacher predictions pre-refinement $p_c(x^i_{j,k})$, but we filter values based on $\tilde{p}_c(x^i_{j,k})$. Consequently, more pixels pass the (unchanged) threshold. We tuned $\alpha_0$ value in Table 6 and set $\alpha_0 = 0.4$, i.e., 60% of raw predictions pass the threshold at $t = 0$.

### 3.3 Putting it All Together

We perform semi-supervised learning for semantic segmentation by pseudo labeling pixels using their neighbors' contextual information. Labeled images and student model prediction used for the supervised loss (2). Unlabeled images are processed by both student and teacher models. We use $\kappa_{\mathrm{margin}}$ (5) of teacher predictions, sort them, and set the threshold $\gamma_t$ ( Section 3.2.2). The per-class teacher predictions are refined using the *weighted union event* relaxation, as defined in Eq. (9). Pixels with top class matching original label and margin values higher than $\gamma_t$ are assigned pseudo labels as described in Eq. (4), for the unsupervised loss (3). The entire pipeline is visualized in Fig. 2.

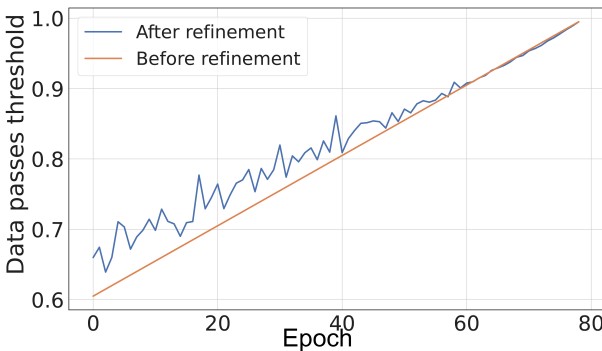 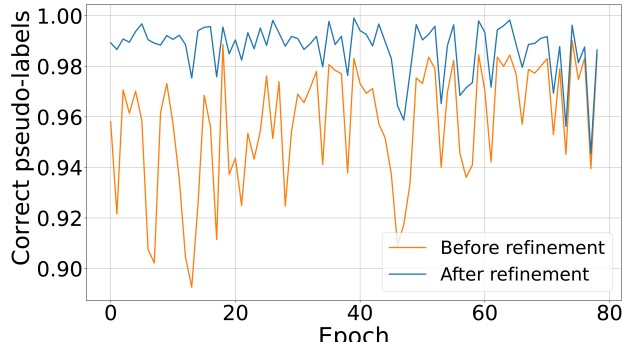

(a) **Data fraction that passes the threshold**. Our method increases the number of pseudo labeled pixels, mostly in the early stage of the training.

(b) **Accuracy of the pseudo labels**. S4MC produces more quality pseudo labels during the training process, most notably at the early stages.

Figure 4: pseudo label quantity and quality on PASCAL VOC 12 (Everingham et al., 2010) with 366 labeled images using our margin (5) confidence function. The training was performed using S4MC; metrics with and without S4MC were calculated.

Table 1: Comparison between our method and prior art on the PASCAL VOC 12 `val` (1,464 original annotated images out of 10,582 in total) under different partition protocols using ResNet-101 backbone. The caption describes the share of the training set used as labeled data and the actual number of labeled images. * denotes reproduced results using official implementation. $\pm$ denotes the standard deviation over three runs.

| Method | 1/16 (92) | 1/8 (183) | 1/4 (366) | 1/2 (732) | Full (1464) |
|---|---|---|---|---|---|
| CutMix-Seg (French et al., 2020) | 52.16 | 63.47 | 69.46 | 73.73 | 76.54 |
| ReCo (Liu et al., 2022a) | 64.80 | 72.0 | 73.10 | 74.70 | - |
| ST++ (Yang et al., 2022b) | 65.2 | 71.0 | 74.6 | 77.3 | 79.1 |
| U$^2$PL (Wang et al., 2022) | 67.98 | 69.15 | 73.66 | 76.16 | 79.49 |
| PS-MT (Liu et al., 2022b) | 65.8 | 69.6 | 76.6 | 78.4 | 80.0 |
| PCR (Xu et al., 2022) | 70.06 | 74.71 | 77.16 | 78.49 | 80.65 |
| FixMatch* (Yang et al., 2023) | 68.07 | 73.72 | 76.38 | 77.97 | 79.97 |
| UniMatch* (Yang et al., 2023) | 73.75 | 75.05 | 77.7 | 79.9 | 80.43 |
| CutMix-Seg + S4MC | 70.96 | 71.69 | 75.41 | 77.73 | 80.58 |
| FixMatch + S4MC | 73.13 | 74.72 | 77.27 | 79.07 | 79.6 |
| UniMatch$^\psi$ + S4MC | **74.72**$\pm$**0.283** | **75.21**$\pm$**0.244** | **79.09**$\pm$**0.183** | **80.12**$\pm$**0.120** | **81.56**$\pm$**0.103** |

The impact of S4MC is shown in Fig. 4, comparing the fraction of pixels that pass the threshold with and without refinement. S4MC uses more unlabeled data during most of the training (a), while the refinement ensures high-quality pseudo labels (b). We further study true positive (TP) and false positive (FP) rates, as shown in Fig. E.2a in the Appendix. We show qualitative results in Fig. 3, including both the confidence heatmap and the pseudo labels with and without the impact of S4MC.

## 4 Experiments

This section presents our experimental results. The setup for the different datasets and partition protocols is detailed in Section 4.1. Section 4.2 compares our method against existing approaches and Section 4.3 provides the ablation study. Implementation details are given in Appendix C.

### 4.1 Setup

**Datasets**  In our experiments, we use PASCAL VOC 12 (Everingham et al., 2010), Cityscapes (Cordts et al., 2016), and MS COCO (Lin et al., 2014) datasets.

Table 2: Comparison between our method and prior art on the PASCAL VOC 12 `val` (1,464 original annotated images out of 10,582 in total) under different partition protocols using ResNet-50 backbone. The caption describes the share of the training set used as labeled data.

| Method | 1/16 (92) | 1/8 (183) | 1/4 (366) | 1/2 (732) | Full (1464) |
|---|---|---|---|---|---|
| Supervised Baseline | 44.0 | 52.3 | 61.7 | 66.7 | 72.9 |
| PseudoSeg (Zou et al., 2021) | 54.89 | 61.88 | 64.85 | 70.42 | 71.00 |
| PC$^2$Seg (Zhong et al., 2021) | 56.9 | 64.6 | 67.6 | 70.9 | 72.3 |
| UniMatch (Yang et al., 2023) | 71.9 | 72.5 | 76.0 | 77.4 | 78.7 |
| UniMatch$^\psi$ + S4MC | **72.62** | **72.83** | **76.44** | **77.83** | **79.41** |

Table 3: Comparison between our method and prior art on the augmented PASCAL VOC 12 `val` dataset under different partitions, utilizing additional unlabeled data from Hariharan et al. (2011) (total of 10,582 training images, 9,118 weakly annotated) and using ResNet-101 backbone. We included the number of labeled images in parentheses for each partition ratio. * denotes reproduced results using official implementation.

| Method | 1/16 (662) | 1/8 (1323) | 1/4 (2646) | 1/2 (5291) |
|---|---|---|---|---|
| CutMix-Seg (French et al., 2020) | 71.66 | 75.51 | 77.33 | 78.21 |
| AEL (Hu et al., 2021) | 77.20 | 77.57 | 78.06 | 80.29 |
| PS-MT (Liu et al., 2022b) | 75.5 | 78.2 | 78.7 | - |
| U$^2$PL (Wang et al., 2022) | 77.21 | 79.01 | 79.3 | 80.50 |
| PCR (Xu et al., 2022) | 78.6 | **80.71** | **80.78** | 80.91 |
| FixMatch* (Yang et al., 2023) | 74.35 | 76.33 | 76.87 | 77.46 |
| UniMatch* (Yang et al., 2023) | 76.6 | 77.0 | 77.32 | 77.9 |
| CutMix-Seg + S4MC | **78.84** | 79.67 | 79.85 | **81.11** |
| FixMatch + S4MC | 75.19 | 76.56 | 77.11 | 78.07 |
| UniMatch$^\psi$ + S4MC | 76.95 | 77.54 | 77.62 | 78.08 |

Table 4: Comparison between our method and prior art on the Cityscapes `val` dataset (total of 2,976 training images) under different partition protocols using ResNet-101 backbone. Labeled and unlabeled images are selected from the Cityscapes `training` dataset. For each partition protocol, the caption gives the share of the training set used as labeled data and the number of labeled images. * denotes reproduced results using official implementation.

| Method | 1/16 (186) | 1/8 (372) | 1/4 (744) | 1/2 (1488) |
|---|---|---|---|---|
| CutMix-Seg (French et al., 2020) | 69.03 | 72.06 | 74.20 | 78.15 |
| AEL (Hu et al., 2021) | 74.45 | 75.55 | 77.48 | 79.01 |
| U$^2$PL (Wang et al., 2022) | 70.30 | 74.37 | 76.47 | 79.05 |
| PS-MT (Liu et al., 2022b) | - | 76.89 | 77.6 | 79.09 |
| PCR (Xu et al., 2022) | 73.41 | 76.31 | 78.4 | 79.11 |
| FixMatch* (Yang et al., 2023) | 74.17 | 76.2 | 77.14 | 78.43 |
| UniMatch* (Yang et al., 2023) | 75.99 | 77.55 | 78.54 | 79.22 |
| CutMix-Seg + S4MC | 75.03 | 77.02 | 78.78 | 78.86 |
| FixMatch + S4MC | 75.2 | 77.61 | 79.04 | 79.74 |
| UniMatch$^\psi$ + S4MC | **77.0** | **77.78** | **79.52** | **79.76** |

**PASCAL** comprises 20 object classes (+ background). 2,913 annotated images are divided into training and validation sets of 1,464 and 1,449 images, respectively. Zoph et al. (2020) shown that joint training of PASCAL with training images with augmented annotations (Hariharan et al., 2011) outperforms joint

training with COCO (Lin et al., 2014) or ImageNet (Russakovsky et al., 2015). Based on this finding, we use extended PASCAL VOC 12 (Hariharan et al., 2011), which includes 9,118 augmented training images, wherein only a subset of pixels are labeled. Following prior art, we conducted two sets of experiments: in the first, we used only subset of the original training data as annotated, while in the second, "augmented" setup, we also used the weakly annotated data and randomly sample images from both.

**Cityscapes** dataset includes urban scenes from 50 cities with 30 classes, of which only 19 are typically used for evaluation (Chen et al., 2018a;b).

**MS COCO** dataset is a challenging segmentation benchmark with 80 object classes (+ background). 123k images are split into 118k and 5k for training and validation.

**Implementation details**   We implement S4MC with teacher–student paradigm of consistency regularization, both with teacher averaging (Tarvainen & Valpola, 2017; French et al., 2020) and augmentation variation (Sohn et al., 2020a; Yang et al., 2023) frameworks. All variations use DeepLabv3+ (Chen et al., 2018b), while for feature extraction, we use ResNet-101 (He et al., 2016) for PASCAL VOC and Cityscapes, and Xception-65 (Chollet, 2016) for MS COCO. For the teacher averaging setup, the teacher parameters $\theta_t$ are updated via an exponential moving average (EMA) of the student parameters: $\theta_t^\eta = \tau \theta_t^{\eta-1} + (1 - \tau)\theta_s^\eta$, where $0 \leq \tau \leq 1$ defines how close the teacher is to the student and $\eta$ denotes the training iteration. We used $\tau = 0.99$. In the augmentation variation approach, pseudo labels are generated through weak augmentations, and optimization is performed using strong augmentations. Additional details are provided in Appendix C.

**Evaluation**   We compare S4MC with state-of-the-art methods and baselines under the standard partition protocols – using 1/2, 1/4, 1/8, and 1/16 of the training set as labeled data. For the "classic" setting of the PASCAL experiment, we additionally use all the finely annotated images. We follow standard protocols and use mean Intersection over Union (mIoU) as our evaluation metric. We use the data split published by Wang et al. (2022) when available to ensure a fair comparison. For the ablation studies, we use PASCAL VOC 12 `val` with 1/4 partition.

### 4.2   Results

Results denoted by * reproduced from Yang et al. (2023) original implementation, resulting in a minor difference in performance. $\psi$ denotes using UniMatch without feature perturbation branch, i.e., only two instances of each image.

**PASCAL VOC 12.**   Tables 1 and 2 compares our method with state-of-the-art baselines on the PASCAL VOC 12 dataset using ResNet-50 and ResNet-101, respectively. Table 3 shows the comparison results on PASCAL with additional unlabeled data from SBD (Hariharan et al., 2011) using ResNet-101. S4MC outperforms all compared methods in standard partition protocols using the PASCAL VOC 12 dataset only, while utilizing SBD, we observe comparable results to the state-of-the-art PCR (Xu et al., 2022). More significant improvement can be observed for partitions of extremely low annotated data, where other methods suffer from starvation due to poor teacher generalization. We observed that using supervised data only yield mIoU of 80.42, so we conducted another experiment, where we used all available annotations from $10,582$ images, and besides, we used the $9,118$ weakly annotated images as unlabeled. This experimental setting utilize the data the most, pushing S4MC to a mIoU of 82.33. Qualitative results are shown in Fig. 3. Our refinement procedure aids in adding falsely filtered pseudo labels and removing erroneous ones.

**Cityscapes.**   Table 4 presents the comparison with state-of-the-art methods on the Cityscapes `val` (Cordts et al., 2016) dataset under various partition protocols. S4MC outperforms the compared methods in most partitions, and combined with the UniMatch scheme, S4MC outperforms compared approaches across all partitions.

**MS COCO.**   Table 5 presents the comparison with state-of-the-art methods on the MS COCO `val` (Lin et al., 2014) dataset. S4MC outperforms the compared state-of-the-art methods in most regimes, using the data splits published in (Yang et al., 2023). In this experimental setting, the model sees a small fraction of

Table 5: Comparison between our method and prior art on COCO (Lin et al., 2014) `val` (total of 118,336 training images) on different partition protocols using Xception-65 backbone. For each partition protocol, the caption gives the share of the training set used as labeled data and the number of labeled images. * denotes reproduced results using official implementation.

| Method | 1/512 (232) | 1/256 (463) | 1/128 (925) | 1/64 (1849) | 1/32 (3697) |
|---|---|---|---|---|---|
| Supervised Baseline | 22.9 | 28.0 | 33.6 | 37.8 | 42.2 |
| PseudoSeg (Zou et al., 2021) | 29.8 | 37.1 | 39.1 | 41.8 | 43.6 |
| PC2Seg (Zhong et al., 2021) | 29.9 | 37.5 | 40.1 | 43.7 | 46.1 |
| UniMatch* (Yang et al., 2023) | 31.9 | 38.9 | **43.86** | 47.8 | 49.8 |
| UniMatch$^\psi$ + S4MC | **32.9** | **40.4** | 43.78 | **47.98** | **50.58** |

Table 6: The effect of $\alpha_0$, the initial proportion of confidence pixels for the Pascal VOC 12 with 1/4 labeled data and ResNet-101 backbone.

| $\alpha_0$ | 20% | 30% | 40% | 50% | 60% |
|---|---|---|---|---|---|
| mIoU | 78.13 | 77.53 | **79.1** | 78.24 | 77.99 |

the data, which could be hard to generalize over all classes. Yet, the mutual information using neighboring predictions seems to compensate somewhat as more supervision signals propagate from the unlabeled data.

**Contextual information at inference.** Given that our margin refinement scheme operates through prediction adjustments, we explored whether it could be employed at inference time to enhance performance. The results reveal no improvement, underlines that the performance advantage of S4MC primarily derives from the adjusted margin, as the most confident class is rarely swapped.

## 4.3 Ablation Study

**Neighborhood size and neighbor selection criterion.** Our prediction refinement scheme employs event-union probability with neighboring pixels. We examine varying neighborhood sizes ($N = 3, 5, 7$), number of neighbors ($k = 1, 2$), and selection criteria for neighbors. We compare the following methods for choosing the neighboring predictions: (a) Random selection, (b) Cosine similarity, (c) Max probability, and (d) Minimal probability from a complete neighborhood. Note that for the cosine similarity, we choose a single most similar prediction vector for all classes, thus mostly enhancing confidence and not overturning predictions. We also compare with $N = 1$ neighborhood, corresponding to not using S4MC. As seen from Table 7, $N = 3$ neighborhood with one neighboring pixel of the highest class probability proved most efficient in our experiments. Aggregating the probability of a randomly selected neighbor has negligible influence on model performance. Using multiple neighbors demonstrates improved performance as the neighborhood expands. Notably, the inclusion of minimal class probability pixels adversely affects model performance, primarily attributed to the contribution of neighbors that exhibit high certainty in belonging to a distinct class.

We also examine the contribution of the proposed pseudo label refinement (PLR) and DPA. Results in Table 8 show that the PLR improves the mask mIoU by 1.09%, while DPA alone harms the performance. This indicates that PLR helps semi-supervised learning mainly because it enforces more spatial dependence on the pseudo labels.

**Threshold parameter tuning** We utilize a dynamic threshold that depends on an initial value, $\alpha_0$. In Table 6, we examine the effect of different initial values to establish this threshold. A smaller $\alpha_0$ propagates too many errors, leading to significant confirmation bias. In contrast, a larger $\alpha_0$ would mask most of the data, rendering the semi-supervised learning process lengthy and inefficient.

Table 7: The effect of neighborhood size and neighbor selection criterion on the Pascal VOC 12 with 1/4 labeled data and ResNet-101 backbone. We denote the number of neighbors as $k$. We compared choosing one neighbor at random, the one with the highest cosine similarity to the pixel embedding, max probable neighbor and min probable neighbor. The idea of similar neighboring pixel is explained in the paper, while comparing to minimum probable neighbor try to see if the spatial information can contradict the prediction, reducing the likelihood to assign pseudo label to the predicted class.

| Selection criterion | Neighborhood size N | | | |
| --- | --- | --- | --- | --- |
| | $1 \times 1$ | $3 \times 3$ | $5 \times 5$ | $7 \times 7$ |
| Random neighbor (k=1) | | 77.03 | 76.85 | 76.87 |
| Cosine similarity (k=1) | | 78.02 | 78.05 | 77.99 |
| Max-prob (k=1) | 77.7 | **79.09** | 78.23 | 77.76 |
| Max-prob (k=2) | | 77.77 | 77.82 | 78.03 |
| Min-prob (k=1) | | 75.62 | 75.11 | 73.95 |

Table 8: Ablation study on the different components of S4MC on top of UniMatch for the augmented Pascal VOC 12 with 1/2 labeled data and ResNet-101 backbone. **PLR** is the pseudo label refinement module and **DPA** is dynamic partition adjustment.

| PLR | DPA | mIoU |
| --- | --- | --- |
| | | 79.51 |
| ✓ | | 79.94 |
| | ✓ | 78.25 |
| ✓ | ✓ | **80.13** |

Table 9: Evaluation of Boundary IoU (Cheng et al., 2021) comparing models trained with UniMatch+S4MC and with FixMatch using 183 (1/1024) annotated images on COCO, both uses Xception-65 backbone as in Table 5.

| FixMatch | FixMatch+S4MC |
| --- | --- |
| **31.1** | 29.9 |

**Mask boundaries** Table 9 demonstrates the limitation of our method in terms of boundary IoU (Cheng et al., 2021). Contrary to the improvement S4MC provides to IoU, the boundary IoU is reduced. That aligns with the qualitative results, as our model predictions masks are smoother in regions far from the boundaries and less confident around the boundaries.

## 5 Conclusion

In this paper, we introduce S4MC, a novel approach for incorporating spatial contextual information in semi-supervised segmentation. This strategy refines confidence levels and enables us to leverage more unlabeled data. S4MC outperforms existing approaches and achieves state-of-the-art results on multiple popular benchmarks under various data partition protocols, such as MS COCO, Cityscapes, and Pascal VOC 12. Despite its effectiveness in lowering the annotation requirement, there are several limitations to using S4MC. First, its reliance on event-union relaxation is applicable only in cases involving spatial coherency. As a result, using our framework for other dense prediction tasks would require an examination of this relaxation's applicability. Furthermore, our method uses a fixed-shape neighborhood without considering the object's structure. It would be interesting to investigate the use of segmented regions to define new neighborhoods; this is a future direction we plan to explore.

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

**Algorithm 1:** Pseudocode: Pseudo label refinement of S4MC, PyTorch-like style.

```
# X: predict prob of unlabeled data B x C x H x W
# k: number of neigbors

#create neighborhood tensor
neigborhood=[]
X = X.unsqueeze(1)
X = torch.nn.functional.pad(X, (1, 1, 1, 1, 0, 0, 0, 0))
for i,j in [(None,-2),(1,-1),(2,None)]:
    for k,l in [(None,-2),(1,-1),(2,None)]:
        if i==k and i==1:
            continue
        neighborhood.append(X[:,:,i:j, k:l])
neighborhood = torch.stack(neighborhood)

#pick k neighbors for union event
ktop_neighbors,neigbor_idx=torch.topk(neighborhood, k=k,axis=0)
for nbr in ktop_neighbors:
    beta = torch.exp((-1/2) * neigbor_idx)
    X = X + beta*nbr - (X*nbr*beta)
```

# A    Visual Results

We present in Figs. A.1 and A.2 an extension of Fig. 3, showing more instances from the unlabeled data and the corresponding pseudo labeled with the baseline model and S4MC. In Fig. A.2 we can see that our method can eliminated undesired entities, while sometimes it also eliminate good predictions, it the process of pseudo-labeling.

Our method can achieve more accurate predictions during the inference phase without refinements. This results in more seamless and continuous predictions, which accurately depict objects spatial configuration.

# B    Computational Cost

Let us denote the image size by $H \times W$ and the number of classes by C.

First, the predicted map of dimension $H \times W \times C$ is stacked with the padded-shifted versions, creating a tensor of shape [n,H,W,C]. K top neighbors are picked via top-k operation and calculate the union event as presented in Eq. (9). (The pseudo label refinement pytorch-like pseudo-code can be obtained in Algorithm 1 for $N = 4$ and $k$ max neighbors.)

The overall space (memory) complexity of the calculation is $O(n \times H \times W \times C)$, which is negligible considering all parameters and gradients of the model. Time complexity adds three tensor operations (stack, topk, and multiplication) over the $H \times W \times C$ tensor, where the multiplication operates k times, which means $O(k \times H \times W \times C)$. This is again negligible for any reasonable number of classes compared to tens of convolutional layers with hundreds of channels.

To verify that, we conducted a training time analysis comparing FixMatch and FixMatch + S4MC over PASCAL with 366 labeled examples, using distributed training with 8 Nvidia RTX 3090 GPUs. FixMatch average epoch takes 28:15 minutes, and FixMatch + S4MC average epoch takes 28:18 minutes, an increase of about 0.2% in runtime.

# C    Implementation Details

All experiments were conducted for 80 training epochs with the stochastic gradient descent (SGD) optimizer with a momentum of 0.9 and learning rate policy of $lr = lr_{\text{base}} \cdot \left(1 - \frac{\text{iter}}{\text{total iter}}\right)^{\text{power}}$.

For the teacher averaging consistency, we apply resize, crop, horizontal flip, GaussianBlur, and with a probability of 0.5, we use Cutmix (Yun et al., 2019) on the unlabeled data.

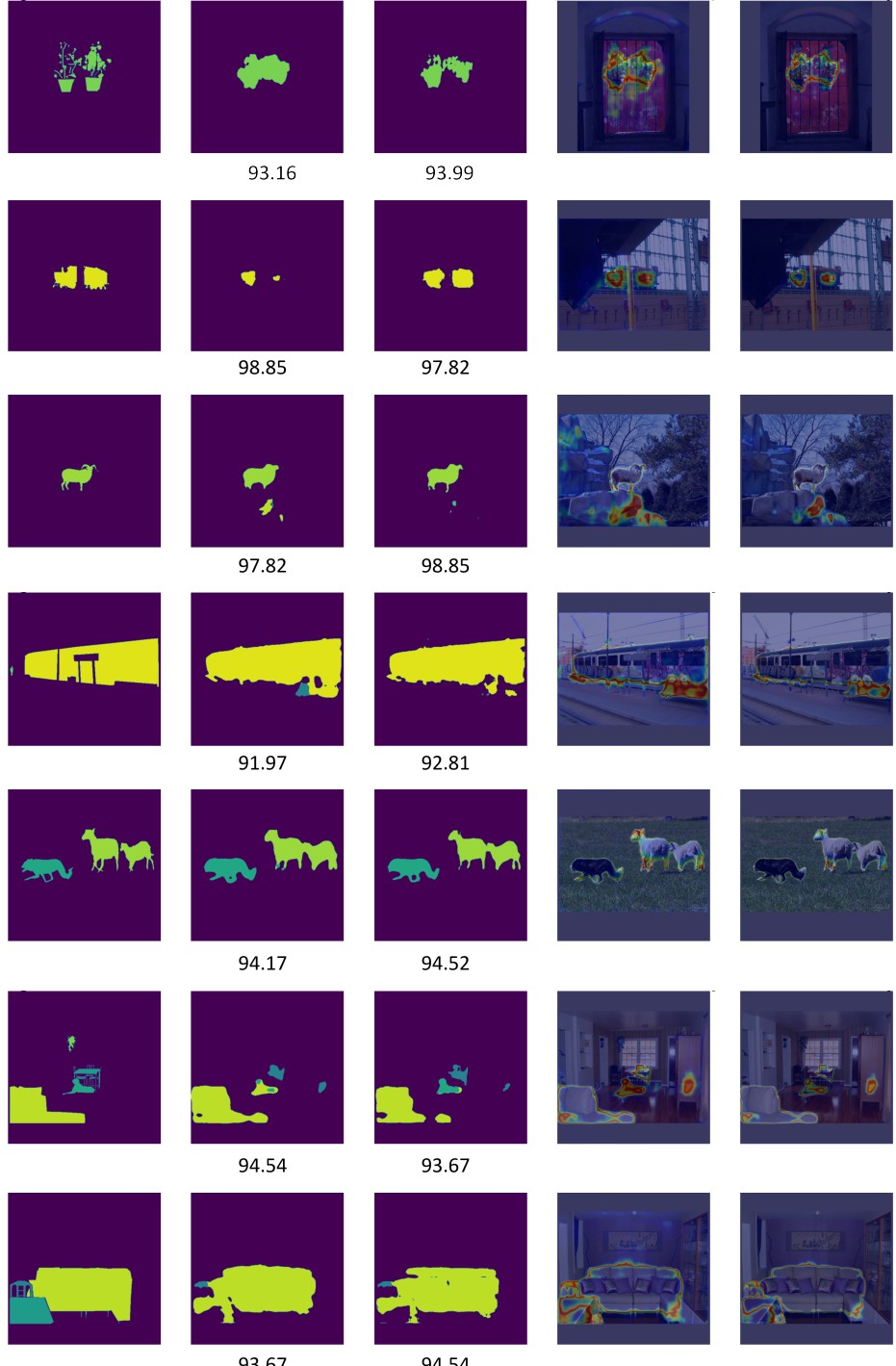

Figure A.1: **Example of refined pseudo labels**, structure of the figure as Fig. 3 and the numbers under the predictions show the pixel-wise accuracy of the prediction map.

For the augmentation variation consistency (Sohn et al., 2020a; Yang et al., 2023), we apply resize, crop, and horizontal flip for weak and strong augmentations as well as ColorJitter, RandomGrayscale, and Cutmix for strong augmentations.

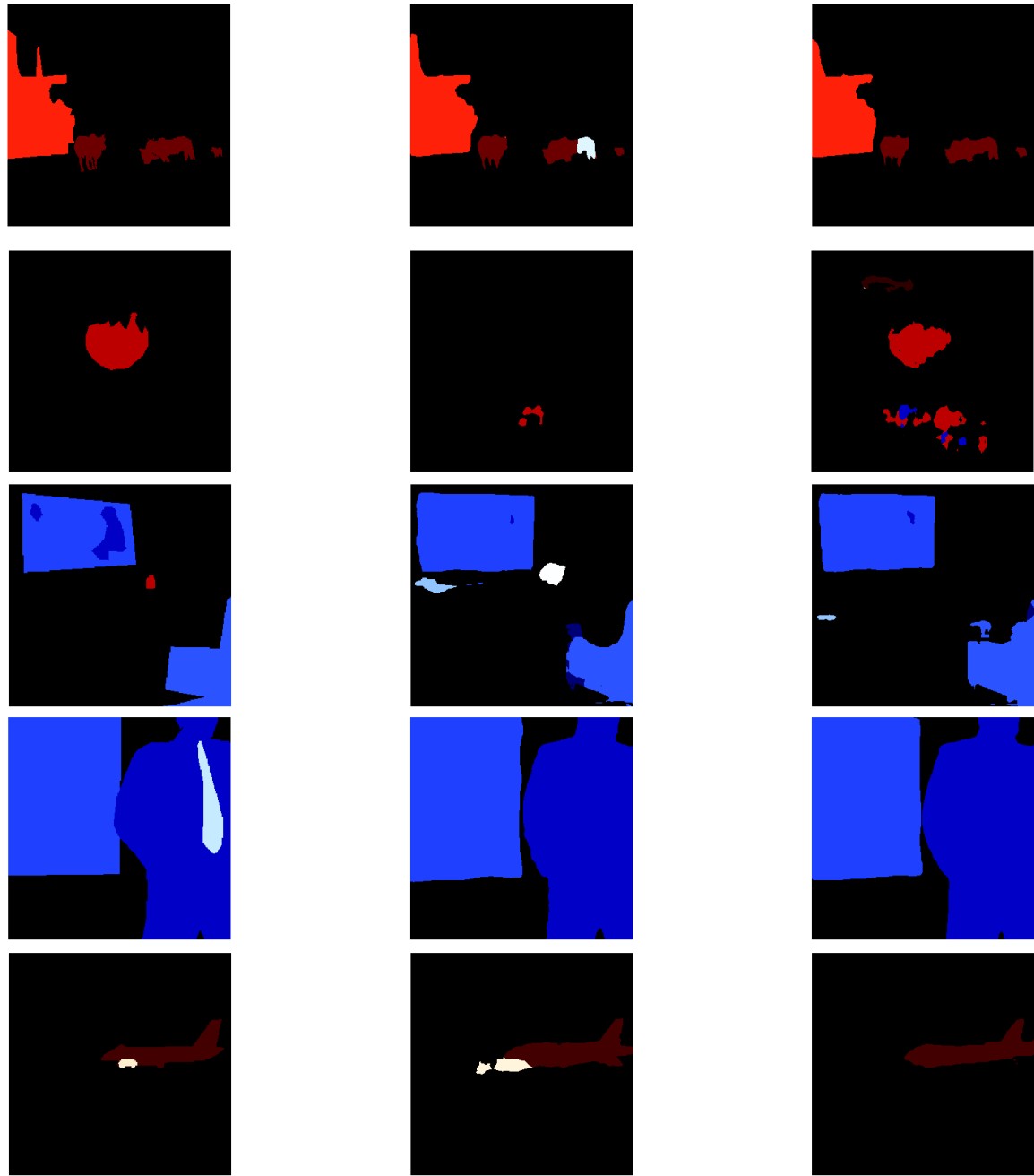

Figure A.2: Qualative results of our method comparison to UniMatch baseline over COCO with 1/32 of the labeled examples. The segmentation map Left to right: Ground Truth, UniMatch prediction, S4MC Prediction

For PASCAL VOC 12 $lr_{\text{base}} = 0.001$ and the decoder only $lr_{\text{base}} = 0.01$, the weight decay is set to 0.0001 and all images are cropped to $513 \times 513$ and $\mathcal{B}_l = \mathcal{B}_u = 3$.

For Cityscapes, all parameters use $lr_{\text{base}} = 0.01$, and the weight decay is set to 0.0005. The learning rate decay parameter is set to power $= 0.9$. Due to memory constraints, all images are cropped to $769 \times 769$ and $\mathcal{B}_\ell = \mathcal{B}_u = 2$. All experiments are conducted on a machine with 8 Nvidia RTX A5000 GPUs.

|  Image | Epoch 10 | Epoch 30 | Epoch 50 | Epoch 70 |

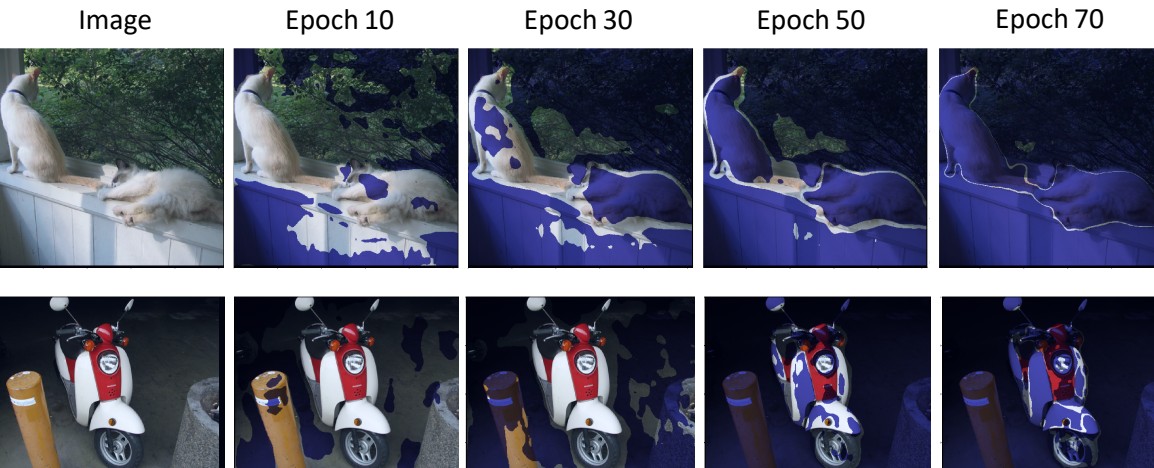

Figure E.1: Qualitative evolution of the pseudo labeling process of S4MC. The figure shows the progress over time of the pixels that assign as pseudo labels w.r.t time.

## D  Limitations and Potential Negative Social Impacts

**Limitations.** The constraint imposed by the spatial coherence assumption also restricts the applicability of this work to dense prediction tasks. Improving pseudo labels' quality for overarching tasks such as classification might necessitate reliance on data distribution and the exploitation of inter-sample relationships. We are currently exploring this avenue of research.

**Societal impact.** Similar to most semi-supervised models, we utilize a small subset of annotated data, which can potentially introduce biases from the data into the model. Further, our PLR module assumes spatial coherence. While that holds for natural images, it may yield adverse effects in other domains, such as medical imaging. It is important to consider these potential impacts before choosing to use our proposed method.

## E  Pseudo Labels Quality Analysis

The quality improvement and the quantity increase of pseudo labels are shown in Fig. 4. Further analysis of the quality improvement of our method is demonstrated in Fig. E.2a by separating the *true positive* and *false positive*, and Fig. E.1 shows qualitative evolution of the pseudo labeling process.

Within the initial phase of the learning process, the enhancement in the quality of pseudo labels can be primarily attributed to the advancement in true positive labels. In our method, the refinement not only facilitates the inclusion of a larger number of pixels surpassing the threshold but also ensures that a significant majority of these pixels are high quality.

As the learning process progresses, most improvements are obtained from a decrease in false positives pseudo labels. This analysis shows that our method effectively minimizes the occurrence of incorrect pseudo labeled, particularly when the threshold is set to a lower value. In other words, our approach reduces confirmation bias from decaying the threshold as the learning process progresses.

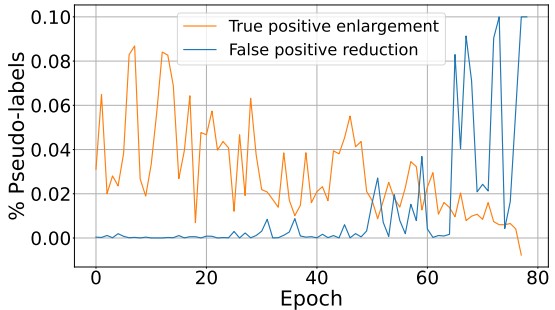
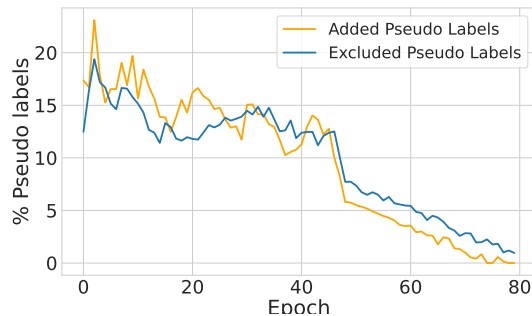

(a) **Quality of pseudo labels** from Fig. 4 separated *True positive* and *False positive* analysis. *True positive* explain the major part of improvement at early stage, while reducing *false positive* explain the enhancement later on.

(b) **Added and excluded pseudo labels**. The pixel-wise pseudo labels S4MC added and excluded over time (i.e. the sum of the graphs is the total pixels that change because of S4MC).

Figure E.2: Analysis of the results of DeepLab V3+ on PASCAL VOC 12

In Fig. E.1 we can see that at late stages of the training process, the bounderies of objects are the most ambiguate and thus the hardest to assign with pseudo labels with S4MC.

## F  Weak–Strong Consistency

We need to redefine the supervision branch to adjust the method to augmentation level consistency framework (Sohn et al., 2020a; Zhang et al., 2021; Wang et al., 2023). Recall that within the teacher averaging framework, we denote $f_{\theta_s}(\mathbf{x}_i)$ and $f_{\theta_t}(\mathbf{x}_i)$ as the predictions made by the student and teacher models for input $\mathbf{x}_i$, where the teacher serves as the source for generating confidence-based pseudo labels. In the context of image-level consistency, both branches differ by augmented versions $\mathbf{x}_i^w$, $\mathbf{x}_i^s$ and share identical weights $f_\theta$. Here, $\mathbf{x}_i^w$ and $\mathbf{x}_i^s$ represent the weak and strong augmented renditions of the input $\mathbf{x}_i$, respectively. Following the framework above, the branch associated with weak augmentation generates the pseudo labels.

### F.1  Confidence Function Alternatives

In this paper, we introduce a confidence function to determine pseudo label propagation. We introduced $\kappa_{\mathrm{margin}}$ and mentioned other alternatives have been examined.

Here, we define several options for the confidence function.

The simplest option is to look at the probability of the dominant class,

$$\kappa_{\max}(x^i_{j,k}) = \max_c p_c(x^i_{j,k}), \tag{F.1}$$

which is commonly used to generate pseudo labels.

The second alternative is negative entropy, defined as

$$\kappa_{\mathrm{ent}}(x^i_{j,k}) = \sum_{c \in C} p_c(x^i_{j,k}) \log\bigl(p_c(x^i_{j,k})\bigr). \tag{F.2}$$

Note that this is indeed a confidence function since high entropy corresponds to high uncertainty, and low entropy corresponds to high confidence.

The third option is for us to define the margin function (Scheffer et al., 2001; Shin et al., 2021) as the difference between the first and second maximal values of the probability vector and also described in the main paper:

$$\kappa_{\mathrm{margin}}(x^i_{j,k}) = \max_c(p_c(x^i_{j,k})) - \mathrm{max2}_c(p_c(x^i_{j,k})), \tag{F.3}$$

Table F.1: Ablation study on the confidence function $\kappa$, over Pascal VOC 12 with partition protocols using ResNet-101 backbone.

| Function | 1/4 (366) | 1/2 (732) | Full (1464) |
|:---:|:---:|:---:|:---:|
| $\kappa_{\text{max}}$ | 74.29 | 76.16 | 79.49 |
| $\kappa_{\text{ent}}$ | 75.18 | 77.55 | 79.89 |
| $\kappa_{\text{margin}}$ | 75.41 | 77.73 | 80.58 |

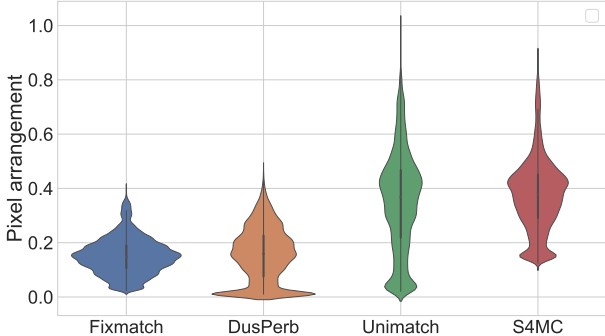 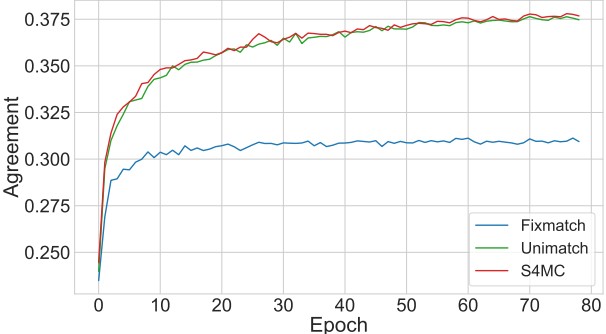

(a) The spatial agreement as we define in in 9 compared between different variations of Unimatch and S4MC.

(b) The spatial agreement, compared between different variations of (Yang et al., 2023) and S4MC over time.

Figure F.1: Spatial agreement analysis off diffrent methods on PASCAL VOC 12 using ResNet-101 backbone.

where max2 denotes the vector's second maximum value. All alternatives are compared in Table F.1.

## F.2 Decomposition and Analysis of Unimatch

Unimatch (Yang et al., 2023) investigating the consistency and suggest using FixMatch (Sohn et al., 2020a) and a strong baseline for semi-supervised semantic segmentation. Moreover, they provide analysis that shows that combining three students for each supervision signal (one feature level augmentation, Channel Dropout, denoted by CD, and two strong augmentations, denoted by S1 and S2) can enhance performance further more, since each student branch can learn a slightly different features. Fusing Unimatch and our method did not provide significant improvements, and we examined the contribution of different components of Unimatch. We measured the pixel agreement as described in Eq. (9) and showed that the feature perturbation branch has the same effect on pixel agreement as S4MC. Fig. F.1 present the distribution of agreement using FixMatch (S1), DusPerb (S1,S2), Unimatch (S1, S2, CD) and S4MC (S1, S2).

## G Bounding the Joint Probability

In this paper, we had the union event estimation with the independence assumption, defined as

$$p_c^1(x_{j,k}^i, x_{\ell,m}^i) \approx p_c(x_{j,k}^i) \cdot p_c(x_{\ell,m}^i) \tag{G.1}$$

In addition to the independence approximation, it is possible to estimate the unconditional expectation of two neighboring pixels belonging to the same class based on labeled data:

$$p_c^2(x_{j,k}^i, x_{\ell,m}^i) = \frac{1}{|\mathcal{N}_l| \cdot H \cdot W \cdot |\mathbf{N}|} \sum_{i \in \mathcal{N}_l} \sum_{j,k \in H \times W} \sum_{\ell,m \in \mathbf{N}_{j,k}} \mathbb{1}\{y_{j,k}^i = y_{\ell,m}^i\}. \tag{G.2}$$

To avoid overestimating that could lead to overconfidence, we set

$$p_c(x_{j,k}^i, x_{\ell,m}^i) = \max(p_c^1(x_{j,k}^i, x_{\ell,m}^i), p_c^2(x_{j,k}^i, x_{\ell,m}^i)) \tag{G.3}$$

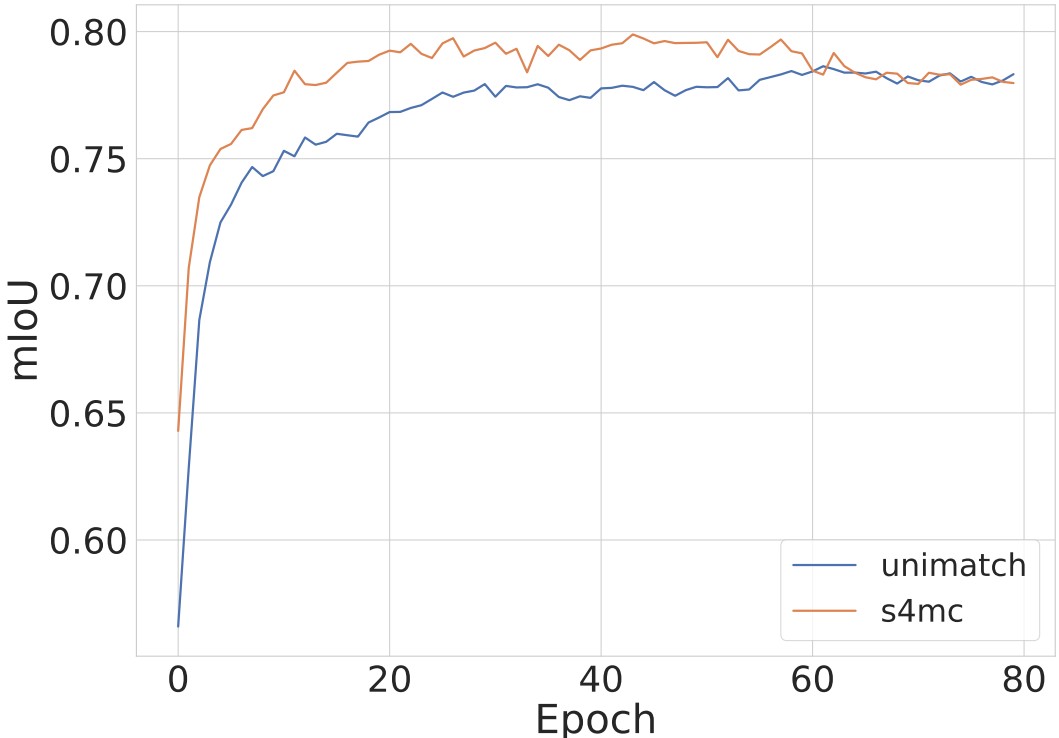

Figure H.1: Training curve of the mIoU of S4MC and Unimatch using Resnet-101 on PASCAL VOC 12 with 366 annotated examples

That upper bound of joint probability ensures that the independence assumption does not underestimate the joint probability, preventing overestimating the union event probability. Using Eq. (G.3) increase the mIoU by **0.22** on average, compared to non use of S4MC refinement, using 366 annotated images from PASCAL VOC 12 Using only Eq. (G.2) reduced the mIoU by **-14.11** compared to the non-use of S4MC refinement and harmed the model capabilities to produce quality pseudo labels.

## H Convergence time

In this section we show that not only S4MC achieve competitive results compared with state-of-the-art methods, but also that PLR helps the model to converge faster. In Fig. H.1 we show that S4MC achieve the same mIoU as Unimatch after only 10 epochs, providing faster convergence in terms of training time using the exact same optimization process.

