# OpenReview forum: "Semi-Supervised Semantic Segmentation via Marginal Contextual Information"
_TMLR — Accepted by TMLR_

### Review · Reviewer_P2oV · 2024-04-13

**Summary Of Contributions:**

The paper presents a modification of pseudo-labeling method for image segmentation problem. Authors propose to revisit the confidence filtering of generated pseudo-labels by refining the probability distribution based on neighbors probability distribution. Doing this allows to have improved quality of pseudo-labels selected and also leads to more data selected every training steps. This results in improvements over the previous methods where the paper’s idea is applied on top. Moreover, authors provide analysis that unimatch prior work actually doing kind of similar thing resulting in only marginal improvement on top of unimatch. Authors also show limitation case of the method like the boundary IoU. The code is available to reproduce / follow the paper.

**Audience:**

Yes

**Broader Impact Concerns:**

I am good with the discussion of limitations and societal risks in appendix D as well as limitation shown in the main body in section 4.3.

**Claims And Evidence:**

No

**Requested Changes:**

**Improvements of text and clarity:**
- be consistent with "pseudo-label" or "pseudo label", Voc 2012 or Voc 12
- be consistent with the sections camel case or lowercase writing (especially in Appendix)
- all Tables need to be fixed for the captions: it is not clear which dataset (2012 or 12 e.g. notation), which model (resnet-51, resnet-100, or other arch as it is different per dataset, better to include), which amount of labeled and unlabeled data are used. Tables 2,3,4,5 also do not have notation of number of labeled examples as stated in the caption.
- link to Hinton - it was not really unlabeled data in the paper, so I would change the reference to something more relevant.
- why is there a link on Arazo 2020 along with Lee 2014? Any reasons to cite Arazo in this context and not other papers?
- Figure 1 - classes which have the max prob is cat then why on the figure it is predicted as background on the left plot?
- Eq. (8) what is $N$?
- "In practice we calculate it iteratively ...." - I don't understand how exactly you compute it
- Sec. 3.2.2 what is $\alpha_0$
- "To determine the DPA threshold, we use ..." - I don't understand this sentence too. How do you determine the threshold itself? Is it just a hyper parameter?
- It is not clear the difference between Table 1 and 2 based on captions.
- Why is there no (full) column in Tables 3,4,5?
- "While Table 2 and 3 ... with unlabeled data ..." - seems Table 2 doesn't use extra unlabeled data.
- Discussion of results that you didn't outperform all methods for all partitions - I suggest that you do an apple-to-apple comparison which is fair to do and you are good to go by saying and for all baselines if you add on the baseline proposed method then it always improves the baseline.
- Sec. 4.2 Cityscapes - it is combined with unimatch not fixmatch!
- Sec. 4.2 "Contextual information at inference" - I don't understand what exactly you do. Also 85.7 and 85.71 are not different, it really seems. What is std?
- Table 6 any caption description to understand what the model, what the data, what the baseline you used? Also max-prob (k=2) it is not clear how it is calculated.
- "Using multiple neighbors demonstrates improved ..." - this is actually opposite from Table 6.
- Change Table 7 and 8 order for better readability and paper flow.
- Table 9 - how much unlabeled data?
- Code in Algorithm 1 (Appendix) - 3 last rows I think are incorrect
- Figure F1 (a) "in in" typo, caption should explain what is going on
- Figure A1 - what is baseline - overall in all places captions and descriptions should be clear.

**Empirical suggestions / questions**
- Figure 4 will be great to extend to the analysis of how many passes that before didn't pass and how many didn't pass that before passed - so the dynamics of how refinement changed the selection itself (not just total number of passes). Also in this graph (b) I think it is missing to show what is happening with PLs accuracy for the baseline. Also in the same (b) figure is it accuracy of the passed PLs or all PLs?
- Table 1 - what is (full)? Is it fully supervised model? How many unlabeled data are used if it is not? why all methods are different with accuracy on in this (full) column if it is fully supervised baseline (all data are labeled)? Why proposed method is worse than the baseline fixmatch if the proposed method is applied to fixmatch in column (full)?
- Improvements on top of unimatch in Table 4 between 1/8 and 1/4 are really very different which is not expecting and surprising for me. Is it std big or any other problem?
- Table 5 - there is no monotonic improvement over the labeled data size growth - why is that?
- Table 6 what happens if 2x2 or 4x4 is used as there is not exact max/min found yet.
- I found it to be strange that having more neighbor pixels of highest class probability is actually hurts a lot (or improvements are marginal compared to one neighbor pixel).
- "PLR improves the task ... while DPA alone harms ..." - this is not true based on Table 8. What is actually correct: text or Table? Also why Table 8 is done on 1/2 and not 1/4 as other ablations?
- There should be at least for one dataset comparison that dynamic confidence threshold over training doesn't give similar value as if authors adds also probability refinement.

**Strengths And Weaknesses:**

**Strengths**
- authors propose a novel and interesting idea, also having a hypothesis of correlation between neighbor pseudo-labels or at least the need of their alignment to have less noisy pseudo-labels.
- extensive experiments showing that the method is working on top of very different prior works and can improve every prior work
- interesting experiment showing the limitation of the method for boundary IoU
- interesting finding what is going on with unimatch: kind of showing that it is doing similar thing as proposed method thus improvements on top are marginal
- diversity of datasets and benchmarks

**Weaknesses**
- The paper is not clearly written in some places, Table captions do not reflect the whole description, inconsistency in the text.
- It is not clear / obvious to me on the choice that at least one pixel in the neighborhood should belong to the class. Why not at least two (or more as a hyper-parameter) should belong to the same class following continuity of the object? This is very not a straightforward choice of the method what authors did while describing the intuition behind, with discussion around neighborhood and taking context into account.
- Authors do not show if their method helps to improve not only quality but also speed of convergence as they are mainly trying to highlight the need of having more reliant data during training and that filtering of many data leads to worse training. Then we really should see (I believe) faster convergence.
- There is absence of fully supervised baseline in all Tables (or it is not clear where it is) to have the upper bound on the performance. Also the dependence of improvements based on the amount of labeled data is kind of strange (it is not monotonic)
- Based on Table 8 it seems that main improvement is coming from the fact that confidence threshold is changing with decay over the training and not because of using refinement of the probabilities. Refinement of the probabilities even make it worse alone and seems adapting the threshold over training is more crucial for it rather than for prior works.

For all weaknesses check details in the requested changes.

---

> ### Author Response · Authors · 2024-04-26
> **Author comment for P2oV review**
>
> We appreciate the reviewer's detailed comments.
>
> ## Text and clarity:
> We embraced the detailed comment and improved all the points raised, improved all captions and mistakes, and added data and backbone to all captions; we would like to address some of the raised points specifically:
>
>
> 1. **Q**: be consistent with terms, sections and add data and architecture:
>
> **A**: Thank you, we corrected everything in the revised version of the paper.
>
> 2. **Q**: link to Hinton
>
> **A**: Thank you. We referred to the basic idea of teacher-student here. We revised it and now cite mean--teacher and future works that use teacher-student for SSL.
>
>
> 3. **Q**: why is there a link to Arazo 2020 along with Lee 2014?
>
> **A**: we have removed the citation.
>
>
> 4.**Q**: Figure 1 - classes which have the max prob is cat, then why on the figure it is predicted as background on the left plot?
>
> **A**: The revised version emphasizes that it illustrates pseudo labels generated solely for the cat class, excluding the background.
>
>
> 5. **Q**: Eq. (8) what is $N$?
>
> **A**: Originally intended to denote the maximum joint probability estimated within an NxN neighborhood, we refine the notation in the revised version with the neighborhood size implied.
>
>
> 6. **Q**: "In practice, we calculate it iteratively ...." I don't understand how exactly you compute it.
>
> **A**: To calculate $p(A\cup B\cup C)$, we first calculate $p(D)=p(A\cup B)$ and then $p(D\cup C)$. We elaborated on this process in the revised version.
>
>
> 7. **Q**: Sec. 3.2.2 What is $\alpha_0$
>
> **A**:  It is the initial value for the DPA; we added the formula in the revised version: $\alpha_t = \alpha_0  (1 - t/iterations)$.
>
>
> 8. **Q**: "To determine the DPA threshold, we use .." How do you determine the threshold itself? Is it just a hyperparameter?
>
> **A**: The threshold is chosen as $\alpha_t$ percentile of pre-refined probabilities $p_c(x^i_{j,k})$, as we define in section 3.1.2 in the revised version.
>
>
> 9. **Q**: difference between Table 1 and 2 not clear based on captions.
>
> **A**: * The difference is the backbone (ResNet-50 and ResNet-101, respectively). We updated the caption accordingly.
>
>
> 10. **Q**:  Why is there no (full) column in Tables 3,4,5?
>
> **A**: For PASCAL VOC 12 experiments, the results reported in Tables 1 and 2 do not utilize weakly annotated data as labeled; as a result, it is possible to use fully labeled data in a semi-supervised setup (with weakly labeled data as unlabeled). This scenario is not available in Tables 3-5.
>
>
> 11. **Q**:"While Table 2 and 3 ... with unlabeled data ..." - seems Table 2 doesn't use extra unlabeled data.
>
> **A**: In the revised version, we refer to the tables with the correct data and correct model. Table 2 uses the same data partition as Table 1.
>
> 12. **Q**: Discussion of results - I suggest that you do an apple-to-apple comparison-  proposed method, then it always improves the baseline.
>
> **A**: Thank you for this observation; we emphasize that in the revised version.
>
>
> 13. **Q**: Sec. 4.2 Cityscapes - it is combined with unimatch, not fixmatch!
> **A**: Corrected in the revised version.
>
>
> 14. **Q**: "Using multiple neighbors demonstrates improvement ..." - this is actually opposite from Table 6.
>
> **A**: We observe here only the use of multiple neighbors with the same neighborhood size, where one may see that increasing the neighborhood size can increase the results when using two neighboring pixels.
>
>
> 15. **Q**: Change Table 7 and 8 order
>
> **A**: Done in the revised version.
>
>
> 16. **Q**: "Contextual information at inference" - 85.7 and 85.71 are not different.
>
> **A**: We corrected it to no improvement.
>
>
> 17. **Q**: Table 6 any caption description to understand the model, data, baseline.
>
> **A**: We elaborate on the comparison between choosing neighbors at random, based on cosine similarity, and selecting the maximum and minimum probable neighbors. While comparing to the minimum probable neighbor to try to see if the spatial information can contradict the prediction, reducing the likelihood of assigning a pseudo label to the predicted class.
>
>
> 18. **Q**: Table 9 - how much unlabeled data?
>
> **A**: 1/1024. It's different from Table 5, but we can run this experiment with a higher percentage of annotated data, so it will match the experiment in Table 5 if it serves the purpose.
>
>
> 19. **Q**: Code in Algorithm 1 (Appendix) - 3 last rows I think are incorrect
>
> **A**: The pseudo-code demonstrates selecting several neighbors in a 3x3 neighborhood, with the last three lines illustrating the process of using more than one neighbor through the associative law of set theory.
>
>
> 20. **Q**: Figure F1 (a) "in in" typo, caption should explain..
>
> **A**: Done in revision.
>
>
> 21. **Q**: Figure A1 - what is baseline - overall captions and descriptions should be clear.
>
> **A**: The baseline is Cutmix-Seg; we fixed the caption in the revised version.

---

> ### Author Response · Authors · 2024-04-26
> **Author comment for P2oV review - part two**
>
> ## Empirical suggestions / questions
>
> 22. **Q**: Figure 4 will be great to extend to the analysis of how many passes that before didn't pass and how many didn't pass that before passed - so the dynamics of how refinement changed the selection itself (not just total number of passes). Also in this graph (b) I think it is missing to show what is happening with PLs accuracy for the baseline. Also in the same (b) figure is it accuracy of the passed PLs or all PLs?
>
> **A**: We will include the extension in the revised version once we’re done; please also consider Figure E1 in the appendix as an extension of where the gain comes from during training.
>
>
> 23. **Q**: Table 1 - what is (full)? Is it fully supervised model? How many unlabeled data are used if it is not? why all methods are different with accuracy on in this (full) column if it is fully supervised baseline (all data are labeled)? Why proposed method is worse than the baseline fixmatch if the proposed method is applied to fixmatch in column (full)?
>
> **A**: We elaborate on using all SBD data as unlabeled and the remaining PASCAL VOC 12 original training data in a revised version.
>
>
> 24. **Q**: Improvements on top of unimatch in Table 4 between 1/8 and 1/4 are really very different which is not expecting and surprising for me. Is it std big or any other problem? Table 5 - there is no monotonic improvement over the labeled data size growth - why is that?
>
> **A**: We regret to say that we do not have insights into why the improvement is not monotonic.
>
>
> 25. **Q**: Table 6 what happens if 2x2 or 4x4 is used as there is not exact max/min found yet.
>
> **A**: We use odd numbers for N, so the neighborhood is symmetric around the refined pixel prediction. If the question is whether there are no surrounding pixels with the same class prediction, this class will gain no additional information from the spatial information and can be reduced if other classes do. Our experiment in Table 9 shows that this happens around the boundaries.
>
>
> 26. **Q**: "PLR improves the task ... while DPA alone harms ..." - this is not true based on Table 8. What is actually correct: text or Table? Also why Table 8 is done on 1/2 and not 1/4 as other ablations?
>
> **A**: The text is correct; the table lines were swapped by mistake. We reordered the numbers in the revised version to accurately reflect this statement.
>
>
> 27. **Q**: There should be at least for one dataset comparison that dynamic confidence threshold over training doesn't give similar value as if authors also probability refinement.
>
> **A**: We have conducted the ablation in T8, and could incorporate additional experiments into Table 1 if deemed beneficial.
>
>
>
> We trust that our response addresses your concerns adequately and are happy to make all the required changes and answer any other concerns that might arise.

---

> > ### Comment · Reviewer_P2oV · 2024-05-07
> > **Thanks for clarifications and revised version**
> >
> > Dear Authors,
> >
> > Sorry for the very late reply (was out for some time). Thank you for all clarifications and detailed responses. Thanks also for revised version.
> >
> > Responses look good to me and now the paper is more clear. I am checking the revised version in details, will come back with comments shortly.
> >
> > Reviewer.

---

### Review · Reviewer_94pr · 2024-04-14

**Summary Of Contributions:**

This paper proposed a confidence margin refinement manner to enhance the semi-supervised semantic segmentation with a teacher-student pipeline. Specifically, for the pseudo-labels produced by the teacher network, the neighboring-based correlation probability is updated before the Dynamic Partition Adjustment (DPA). Thus the proposed S4MC could achieve superior confidence filtering and semi-supervised segmentation performance. Experiments on PASCAL VOC12, COCO, and Cityscapes show the effectiveness of S4MC.

**Audience:**

Yes

**Broader Impact Concerns:**

The authors have discussed the broader impact in Appendix D.

**Claims And Evidence:**

Yes

**Requested Changes:**

Weaknesses 1 and 2 should be further clarified, which are both important to be considered for acceptance.

**Strengths And Weaknesses:**

Strengths:
1. The paper is well-written, while the proposed confidence refinement is easy to implement as clearly mentioned in Sec3.2 and Algorithm1.
2. The authors have discussed the computational cost in Appendix B.
3. Convincing results are shown in Fig.4. Good quantitative experimental results.

Weakness:
1. The main concern is the contradictory presentation of the confidence refinement discussed in the paper. As shown in Eq(3) and Eq(4), only the argmax class of pseudo label influence the $\mathcal{L}_u$. So it is confusing why S4MC still works while "S4MC primarily derives from the adjusted margin, as the most confident class is rarely swapped" as mentioned in Sec4.2?
2. No qualitative results are compared in the paper, while only some confidence refinement examples are shown here.
3. More details should be discussed about Fig.4. For example, why the fraction is consistent with/without refinement when the data passing threshold is high (>0.9).
4. Some minor problems: FP was defined twice for Feature perturbations and False positive. The data range should be [0,100] rather than [0,1.0] with symbol "%" in Fig.4.

---

> ### Author Response · Authors · 2024-04-26
> **Author comment for 94pr review**
>
> We would like to thank you for the hard work of the review, we hope that we adress your concerns properly in the following comment.
>
> 1. **Q**: The main concern is the contradictory presentation of the confidence refinement discussed in the paper. As shown in Eq(3) and Eq(4), only the argmax class of pseudo label influence the L_u . So it is confusing why S4MC still works while "S4MC primarily derives from the adjusted margin, as the most confident class is rarely swapped" as mentioned in Sec4.2?
>
>
> **A**: Indeed only the most confident class affects the pseudo label assignment, but since we refine all classes, they do swap during the training. The statement about rarely swapping classes refers to the use of the refinement mechanism as a post-training component (similar to CRF). Important to note that during training, still the more common phenomenon is not class swapping, but making the most dominant class more confidant, and thus creating more supervision signals earlier in the training process.
>
>
> 2. **Q**: No qualitative results are compared in the paper, while only some confidence refinement examples are shown here.
>
> ** A **: Figures 3, A1, and A2 show qualitative results over the PASCAL and COCO datasets. All the figures show the results after training the model, i.e., final predictions. Once the experiments are done, the revised version will include qualitative results of the pseudo-labeling, as shown in Figures 1 and 2.
>
>
> 3. **Q**: More details should be discussed about Fig.4. For example, why the fraction is consistent with/without refinement when the data passing threshold is high (>0.9).
>
> **A**: The nature of PLR is to use spatial information and make more precise pixel predictions. When the threshold decays so most pixels exceed it (>0.9), the PLR changes are not significant (as an example, the threshold can be 0.3, prediction before refinement can be 0.5 and after refinement 0.55).
>
>
> 4. **Q**: Some minor problems: FP was defined twice for Feature perturbations and False positive. The data range should be [0,100] rather than [0,1.0] with symbol "%" in Fig.4.
>
> **A**: Thank you, we fixed all those issues in the revised version.
>
> Please let us know if any other question might arise.

---

> > ### Comment · Reviewer_94pr · 2024-04-27
> > **Thanks for the response.**
> >
> > Thanks for the response from the authors.
> > I am still confused for the question 1. Simply making the dominant class more confident should not influence the objectives of Eq(3) and Eq(4), because only the argmax label is used for semi-supervised learning rather than confidence.

---

> ### Author Response · Authors · 2024-04-27
> **assigning pseudo label clarification**
>
> The pseudo-label $\hat{y}$ (defined in Eq(4) and appearing in Eq(3)) is indeed a one-hot pseudo-label and is not directly affected by the confidence. Instead, it is determined by the margin between the confidences of the two most dominant classes, as proposed in Eq(5). Consequently, if a new enhanced prediction surpasses the threshold, it assigns a pseudo-label where it did not before.
>
> Since we enhance all classes, both the score of $\max_c$ and ${\mathrm{max2}}_c$ change, and thus the pseudo-label as well.
>
> We should and would be better to clarify that we use $\kappa_{margin}$ with the enhanced prediction vector $\tilde{p}$ from Eq(9) for pseudo-label assignment.

---

> > ### Comment · Reviewer_94pr · 2024-04-28
> > **Thanks for the reply.**
> >
> > Thanks for the detailed explanation. Now I understand the effect of the proposed method on $\kappa_{margin}$.

---

> ### Comment · Reviewer_94pr · 2024-05-08
> **Response for the revision**
>
> Thanks for the rebuttal and the revision. They address most of my concerns. However, I still think this paper needs more qualitative comparison with other methods (such as some approaches compared in Tab1, Tab3) rather than comparing to the baseline only (Fig3, FigA.1, FigA. 2), which would give more intuitive insights.

---

> > ### Author Response · Authors · 2024-05-08
> > **comment on 94pr on qualitative results**
> >
> > Please note that Fig A.2 compare results of our method and Unimatch, the best competitive work in COCO val set.
> >
> > We belive that since our method is combined with existing methods, it gives more insights to see the benefit of our contribution on top of other methods.
> >
> > If you still think it's nececary, the best competitive work on PASCAL is still Unimatch for the majority part of the partitions and models, we can add qualitative results compared with it, similar to Fig A.2.

---

> > > ### Comment · Reviewer_94pr · 2024-05-08
> > > **Thanks**
> > >
> > > Thanks for the reply. It would be better to update the tag from "baseline" to specific methods for these figures.

---

> ### Author Response · Authors · 2024-05-23
> **Replay**
>
> We have deleted the headline of the figure and took your suggestion and added it to the figure description both in Appendix A1, A2 and the main paper Figure 3

---

### Review · Reviewer_KSiJ · 2024-04-23

**Summary Of Contributions:**

This paper presents a technique for semi-supervised learning of semantic segmentation, based on pseudo-labelling. The setup is fairly standard: student-teacher, with teacher being EMA of the student, and pseudo-labels are obtained by checking if a pixel's relative confidence exceeds an adaptive threshold. The key idea in this work is to expand the set of estimates considered "confident" by checking if an estimate has support from its spatial neighbors. By replacing single-pixel confidence with a max-pooled confidence (I am approximating some details of Eqs. 6-8 here), more "good" pixels are let through the thresholding. This leads to a ~1 point improvement in accuracy, which is small, but this is a consistent improvement across different datasets and task settings.

**Audience:**

Yes

**Claims And Evidence:**

Yes

**Requested Changes:**

I also have several small gripes about the notation and about the method description, which I expect can be resolved easily.

An image is \textbf{x}_i, and a pixel is \textbf{x}_{j,k}^i,  but then later pixels are sometimes {x}^i_{j,k} (not bold). Two issues here: (1) whether pixels are bold or not, this should be consistent; (2) superscript/subscript for the image index should be consistent, so making images \textbf{x}^i would probably be best.

Above equation 6, the discussion about "two pixels" is confusing and I'm not sure it makes sense. The idea of a neighborhood is introduced, and sounds promising, and then suddenly the text is talking about 2 pixels, and there is no clear connection.

The max in equation 8 does not seem to be maxing over the right things -- it is a max over $l$ and $m$, but why? The surrounding text suggests it should be a max over the neighborhood. Also, the superscript \textbf{N} is never properly introduced.

In section 3.3, it says "Labeled images are only fed into the student model, producing the supervised loss(2). Unlabeled images are fed into the student and teacher models." I am quite sure this is not true. My understanding is that labelled and unlabelled images are fed into the student, and only unlabelled are fed into the teacher. The teacher is, after all, only an EMA of the student.

The word "Epoch" in Fig4a is cut off.

**Strengths And Weaknesses:**

*Strengths*

This paper is well written, the idea is well motivated, and the key ideas are simple and clear.

The coverage of related work looks fair.

The figures are informative and helpful.

The consistent improvement in results is convincing.



*Weaknesses*

The main technical weakness that jumps out at me is the threshold parameter tuning. I believe that prior work is weak in this respect as well, so this is not a critical problem, but still it makes me nervous. I would love to see some assurance that this sort of tuning is worked out on a validation set (never looking at test results), and maybe even done so on a single dataset, and then the choice is transferred to other datasets with good effect.

---

> ### Author Response · Authors · 2024-04-26
> **Author comment for KSiJ review**
>
> We would like to thank you for investing your time in our work, and we hope that the answers below will address any concerns that have arisen from the paper.
>
> 1. **Q**: The main technical weakness that jumps out at me is the threshold parameter tuning. I believe that prior work is weak in this respect as well, so this is not a critical problem, but still it makes me nervous. I would love to see some assurance that this sort of tuning is worked out on a validation set (never looking at test results), and maybe even done so on a single dataset, and then the choice is transferred to other datasets with good effect.
>
> **A**: We ablate the use of the DPA with different values of $\alpha$ in Table 6 for one Dataset (PASCAL) and use the same value for Cityscapes and COCO. It is possible that we did not utilize the most out of those Datasets.
>
>
> 2. **Q**: An image is \textbf{x}_i, and a pixel is \textbf{x}{j,k}^i, but then later pixels are sometimes ${x}^i_{j,k}$ (not bold). Two issues here: (1) whether pixels are bold or not, this should be consistent; (2) superscript/subscript for the image index should be consistent, so making images \textbf{x}^i would probably be best.
>
> **A**: Thank you, we fixed the notation to be consistent. Note that in the revised version, when we refer to an image, we denote it by $x_i$ (bold) and a specific pixel denoted by $x^i_{j,k}$ (not bold).
>
>
> 3. **Q**: Above equation 6, the discussion about "two pixels" is confusing and I'm not sure it makes sense. The idea of a neighborhood is introduced, and sounds promising, and then suddenly the text is talking about 2 pixels, and there is no clear connection.
>
> **A**: Our method looks at a neighborhood and selects the most significant neighbors for each class. If we chose one most significant neighbor, it is indeed one pixel per class. The example of one neighbor is the simplest to understand. We state that this is one example and explain how to use more than one neighbor in section 3.2.1 in the revised version.
>
>
> 4. **Q**:The max in equation 8 does not seem to be maxing over the right things -- it is a max over 𝑙 and 𝑚, but why? The surrounding text suggests it should be a max over the neighborhood. Also, the superscript \textbf{N} is never properly introduced.
>
>
> **A**: The max is indeed over the neighborhood. N originally intended to denote the maximum joint probability estimated within an NxN neighborhood, we refine the notation in the revised version with the neighborhood size implied.
>
> 5. **Q**: In section 3.3, it says "Labeled images are only fed into the student model, producing the supervised loss(2). Unlabeled images are fed into the student and teacher models." I am quite sure this is not true. My understanding is that labelled and unlabelled images are fed into the student, and only unlabelled are fed into the teacher. The teacher is, after all, only an EMA of the student.
>
> **A**: You are correct, that is what we meant here, but we separate labeled and unlabeled images, as also described in Figure 2. Please let us know if it remains unclear.
>
>
> 6. **Q**: The word "Epoch" in Fig4a is cut off.
>
> **A**: fixed in the revised version.

---

> > ### Comment · Reviewer_KSiJ · 2024-05-03
> >
> > It's too late to do this now, but it would have been great if the revisions were in an alternate color.
> >
> > 1. Table 6 is not what I'm asking for, but your answer here is indeed satisfactory. You should add that to the paper.
> >
> > 2. Great, but it still seems like superscript and subscript are getting mixed up sometimes.
> >
> > 3. OK reads better now.
> >
> > 4. Great, looks fixed.
> >
> > 5. Hm OK, if my summary of the method is correct, then you should fix the weird sentence I quoted.
> >
> > 6. Great.

---

> ### Author Response · Authors · 2024-05-05
> **Reply**
>
> We apriciate your comment and uploaded a revised version with some requsted changes.
>
> We made changes of notation to be consistent and changed the sentence quated at 5. We will carfully look over all superscripts and subscripts to avoid confusions, if still exist after our changes so no mistakes will remain.

---

### Author Response · Authors · 2024-04-26
**Revised version and reviews comments**

Dear reviewers,

First, we want to thank you for your bright insights that helped us improve our manuscript.

We addressed your questions individually and uploaded a revised version with the following changes:

* More consistent with sections, datasets, and other terms

* Fixed typos

* Changed citations to be more accurate

* Emphasizes Figure 1 illustrates pseudo labels generated solely for the cat class.

* Explicitly describe $N$ as the pixel neighborhood and exclude it from Eq 8.

* Rephrasing the use of  multiple neighbors (last paragraph of 3.2.1 and Algorithm 1 last part)

* Added explicit definition of $\alpha_t$

* Better captions of all tables, with the backbone used, and explicit number of images for each partition.

* Changed the order of Tables 7 and 8

* Fixed the order to match the real results in Table 7 (used to be Table 8).

* FP stands for False Positive, in which the mechanism of feature perturbation is changed to channel dropout, as described in the relevant work.

* Consistency with the order of subscripts and superscripts.

* Stating before Eq 6 that this is an example of using one neighbor.

* Improved version of Fig4 with no % and no cut offs.



The revised version does not possess qualitative results of pseudo labels during the training process. It lacks a comparison of the number of pseudo labels we include versus those in a baseline method, whether included or excluded from the baseline during the training process, regardless of correctness.

Both aspects raised by the reviewers are currently in progress and will be uploaded to a revised version once completed.

Thank you again for your time. Please let us know if you are not satisfied with any of the changes.

---

### Author Response · Authors · 2024-05-02
**Revised version of supplementary**

Dear reviewers,

We would like to bring to your attention that we have added a revised version of the supplementary material that includes the Qualitative evolution of the pseudo-labeling process (Figure E.1), as well as another decomposition analysis of Figure 4 into pixel-level added and excluded pseudo-labels (Figure E.2 (b)).

Regards,
The Authors

---

> ### Comment · Reviewer_P2oV · 2024-05-07
> **Feedback on the revised version**
>
> Dear authors,
>
> Thanks again for incorporating all the feedback. Minor comments:
> - Caption Fig 1: "Confidence refinement. observation over " -> remove dot
> - Sec 2.1 beginning "pseudo labeling" - Capitalize the first word in the sentence "Pseudo labeling"
> - In Table 3, does it mean that you take all 10,582 (mix of labeled and weakly labeled data) and sample from them say 1/2 (5291) images as labeled and the rest are used as unlabeled? Then for Table 3 you could train on labeled + weakly labeled as fully supervised baseline to have the upper bound on the performance.
> - "Discussion of results - I suggest that you do an apple-to-apple comparison- proposed method, then it always improves the baseline." I still feel you can do more stress in Tables discussions by comparing X algorithm with X + PLR + DPA results. But I am ok with the revision as is too.
> - About Table 8 and extra results that DPA is not bringing much - I am good as you fixed the Table and now results are consistent with the text and you showed here that DPA is not solving the issues, which PLR does. :) thanks for fixing Table removing misleading results.
> - Fig E.1. "S4MC. the figure show" - "The" - capitalize and also "shows"
> - Fig E.2 (b) - dot is missing at the end of (b) and the main figure caption. "Compared" -> "Comparison between"? for (a) "labels **from** Fig.4"?
> - For Fig E2 (b) - does it mean that in average you approximately have the same number of samples to train on (as added and excluded closely follow each other) but you have very different data which are used with pseudo-labels? Kind of your method selects differently?
> - F2 "that shows that combining " - there is unfinished English sentence, not clear what they show.
>
> Could you comment still on my two main questions:
> - It is not clear / obvious to me on the choice that at least one pixel in the neighborhood should belong to the class. Why not at least two (or more as a hyper-parameter) should belong to the same class following continuity of the object? This is very not a straightforward choice of the method what authors did while describing the intuition behind, with discussion around neighborhood and taking context into account.
> - Authors do not show if their method helps to improve not only quality but also speed of convergence as they are mainly trying to highlight the need of having more reliant data during training and that filtering of many data leads to worse training. Then we really should see (I believe) faster convergence.
>
> For the second I wonder if you have just training curves and you see that you need less number of training steps to converge compared to the baselines. For the first - maybe you commented on it and I missed it; also ablation in Table 7 of the revised version (where k=2) is showing this (I guess?), but then we need to have bigger neighborhood which maybe be not computationally attractive.
>
> Apart from the above, I think the revised version reads better and I am good with this version.
>
> Reviewer.

---

> > ### Author Response · Authors · 2024-05-08
> > **Authors comment**
> >
> > We want to thank you again for the attention you give to all the details we missed.
> >
> > We corrected all the mistakes and updated the version once more.
> >
> > * Table 3: the answer is yes and we could run this experiment and report here the results in few days.
> >
> > * For Fig E2 (b): Your observation is correct. We do produce more labels in early stages (at the most it is about 5%) and slightly less in later stages, yet as fig E2b suggest (and Fig 4b validates) the main contribution comes from the quality of the pseudo-labels and not from increase in quanttity.
> >
> >  *F2 "that shows that combining": We completed and made more readable. The gaol of their work is that it can enhance performance further more, since each student branch can learn a slightly different features.
> >
> > * Q1: The choice that at least one pixel in the neighborhood-
> > * A1: We demonstrate through the use of one neighbor, but it is entirely possible to use more than that. Empirically, it shows that choosing one is more sufficient, since using more neighbors can further confuse the model.
> > By looking at Boundary IoU, we believe that employing more neighbors along the boundaries confuses our method even more. We believe that your intuition is correct, and one can use an adaptive number of neighbors and neighborhood size, perhaps based on the absolute confidence of the model and the scale of the instance. Larger instances and convex instances might perform better with larger neighborhoods and more neighboring predictions. We have not come across a better recipe so far and leave that for future work.
> >
> > * Q2: convergence time.
> > * A2:  You are correct, we did not thing that it is a sagnificant point to present in the paper, we've added section H and a figure in appendix H.

---

> > > ### Comment · Reviewer_P2oV · 2024-05-08
> > > **Reply**
> > >
> > > Thanks for added convergence time! this looks nice, and even we can see that we start overfit if we train way longer as unimatch. It definitely shows convergence speed up. Just cosmetic: add dot at the end of caption for H.1 fig and specify if this is train mIoU or validation.
> > >
> > > About neighbors: thanks for the clarification and yep, agree - deeper investigation (as another paper) is needed to account boundary effects.
> > >
> > > Thanks again for all work and discussion! Just wait last number on fully supervised baseline for Table 3 - with the rest I am good!

---

> > > > ### Author Response · Authors · 2024-05-12
> > > > **Fully supervision on Table 3**
> > > >
> > > > We have made two setups for the fully supervision, since the major part of the data has reduced annotations.
> > > >
> > > > * First experiment uses no pseudo-labeling, simply supervised learning for what has annotations. This lead to 80.42 mIoU.
> > > >
> > > > * Second experiment was conducted after getting the first results, understanding that the weakly annotated data contribute less, even when using it all. For that experiment we used all the annotations, but we also used all the weakly annotated data as unlabeled data to produce pseudo labels as well. In this setup, we got 82.33 mIoU.
> > > >
> > > > It is important to say that for the second experiment, other methods might differ, so we can't say it is the upper bound for Semi supervised semantic segmentation with all data, but it is the closest to the title "fully utilizing" the data.
> > > >
> > > > Since other methods did not run such an experiment, we incorporated this experiment in the text and not in the table, as you will see in the revised version we will upload today

---

> > > > > ### Comment · Reviewer_P2oV · 2024-05-13
> > > > > **Reply about Table 3**
> > > > >
> > > > > Dear Authors,
> > > > >
> > > > > Thanks for that baseline. Right I understand that this 80.42 supervised baseline is the lower bound also for Table 1 last column as it is trained on 1,464 labeled images? For the second experiment, what did you get when use 1,464 labeled + 9,118 weakly labeled but treated as labeled? I presume issue that weakly labels are too noisy, thus better to use them less.

---

> > > > > > ### Author Response · Authors · 2024-05-13
> > > > > > **replay supervision numbers**
> > > > > >
> > > > > > Dear reviewer,
> > > > > >
> > > > > > We want to make it more clear.
> > > > > >
> > > > > > Training on PASCAL the DeepLab V3+ (resnet101),
> > > > > > * 1,464 original images supervision - 77.92
> > > > > > * 10,582 total images supervision - 80.42
> > > > > > both of this numbers should be similar for all other methods that uses the same model and same input scaling (retulst might vary by a small change due to training procedure).
> > > > > > In table 1 (and 2 and 3) we use all the images, as images that is not picked for supervision used as unlabeled data. The main diffrence is that in table 1 we choose labeled images only out of the fine annotated ones, while table 3 images choosen randomly out of all images, leading to choosing some images with very limited annotations.
> > > > > > For more insights about the limitation of our method, we conducted an experiment that uses all  10,582 images as labeled data, but we've used the 9,118 "augmented" images also as unlabled, trying to find an upper bound to our method. This experiment is not comparable to other methods, and yeild a results of 82.33.

---

> > > > > > > ### Comment · Reviewer_P2oV · 2024-05-13
> > > > > > > **Reply**
> > > > > > >
> > > > > > > Thanks! Now I got it! This is super useful points!

---

### Decision · Action_Editor_MFJK · 2024-05-24

**Recommendation:** Accept as is

**Comment:**

All the reviewers are positive about the paper, citing extensive empirical evaluation and quality of the presentation. All the major reviewers' comments were already incorporated into the manuscript. Thus, I concur and recommend the paper for acceptance without extra revisions.

**Audience:**

Yes, the paper is relevant for TMLR's audience.

**Claims And Evidence:**

Yes, the paper presents a convincing and thorough experimental evaluation that supports all the main claims.